# CAUSALLY MOTIVATED DIFFUSION SAMPLING FRAMEWORKS FOR HARNESSING CONTEXTUAL BIAS

## ABSTRACT

Diffusion models have shown remarkable performance in text-guided image generation when trained on large-scale datasets, usually collected from the Internet. These large-scale datasets have contextual biases (e.g., co-occurrence of objects) which will naturally cascade into the diffusion model. For example, given a text prompt of "a photo of the living room", diffusion models frequently generate a couch, a rug, and a lamp together while rarely generating objects that do not commonly occur in a living room. Intuitively, contextual bias can be helpful because it naturally draws the scene even without detailed information (i.e., visual autofill). On the other hand, contextual bias can limit the diversity of generated images (e.g., diverse object combinations) to focus on common image compositions. To have the best of both worlds, we argue that contextual bias needs to be strengthened or weakened depending on the situation. Previous causally-motivated studies have tried to deal with such issues by analyzing confounders (i.e., contextual bias) and augmenting training data or designing their models to directly learn the interventional distribution. However, due to the large-scale nature of these models, obtaining and analyzing the data or training the huge model from scratch is beyond reach in practice. To tackle this problem, we propose two novel frameworks for strengthening or weakening the contextual bias of pretrained diffusion models without training any parameters or accessing training data. Briefly, we first propose causal graphs to explicitly model contextual bias in the generation process. We then sample the hidden confounder due to contextual bias by sampling from a chain of pretrained large-scale models. Finally, we use samples from the confounder to strengthen or weaken the contextual bias based on methods from causal inference. Experiment results show that our proposed methods are effective in generating more realistic and diverse images than the regular sampling method.

## 1 INTRODUCTION

Diffusion models (Sohl-Dickstein et al., 2015; Ho et al., 2020) have shown remarkable performance in image generation regarding realism (Dhariwal & Nichol, 2021), likelihood estimation (Nichol & Dhariwal, 2021), and controllability (Zhang et al., 2023b; Ruiz et al., 2023; Gal et al., 2023; Hertz et al., 2023), which related research fields have leveraged to provide significant gains in performance, such as Video Generation (Ho et al., 2022b;a), Text-to-3D Synthesis (Poole et al., 2023; Wang et al., 2023), Medical Domain (Kazerouni et al., 2023), and Virtual Try-on (Zhu et al., 2023).

Aside from the sophisticated formulations and optimized modeling techniques, one of the most fundamental reasons for the recent success of diffusion models is the large-scale training data. Large-scale data contains a massive amount of knowledge in the visual world, which could enable the models to learn more extended support and smoother latent space. Billions of images are used to train StableDiffusion (Rombach et al., 2022), an off-the-shelf diffusion model, with which surprisingly realistic images can be generated given diverse user-specified queries.

Despite the superior performance in image generation, pretrained diffusion models implicitly learn contextual bias (e.g., co-occurring objects) from the real-world training data. For example, a set of concepts "a living room", "a couch", "a rug" and "a lamp" are highly correlated in the real world, and thus generated images conditioned on "a living room" frequently contain all of the others, e.g., Fig. 1 (a). We hypothesize this is caused by contextual bias contained in data (Liu et al., 2022; Wang

et al., 2020; Sedikides et al., 1998; Egglin & Feinstein, 1996), which naturally cascades into large models during the training.

While this contextual bias is widely observed, is contextual bias inherently bad? Not necessarily. Contextual bias represents how the training data statistically looks like, albeit a possibly biased sample of the real world. For instance, some objects are frequently placed in the living room together, and thus a living room without all the co-occurring objects may not look like a living room anymore. In other words, contextual bias could be an important ingredient in generating a natural scene.

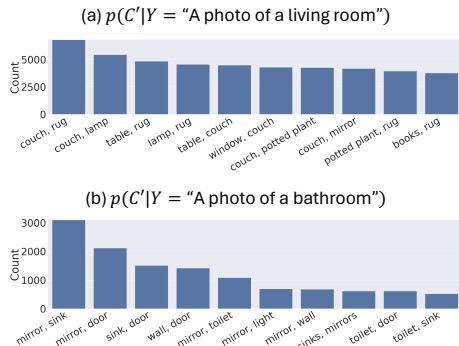

However, this does not necessarily mean that contextual bias is always good because it may reflect the distribution of photos online rather than the real visual world. In reality, an object is not always placed with its co-occurring objects. It can be placed with some objects not correlated, or it can be placed without some objects highly correlated. For example, there might be a living room with kitchenware, or there might be a living room

Figure 1: Visualizations of contextual biases for given scene descriptions. 10,000 generated samples are used for counting object co-occurrence.

without a couch or a lamp. This means that contextual bias could limit the spectrum of generated objects to frequently co-occurring objects while degenerating the diversity of the generated objects.

To have the best of both worlds, we believe that contextual bias needs to be explicitly modeled to be controlled in the generation process. In practice, strengthening and weakening contextual bias can be useful in multiple scenarios. First, if a given condition is not detailed enough to describe the whole scene, the generated sample can miss some objects that should have been put together. This is not an unusual scenario, considering that (1) a caption of an image mostly describes only a part of the image[1] (Lin et al., 2014; Krishna et al., 2017), and (2) a simple interaction (e.g., short prompt) can be preferred by end-users in general (Krug, 2000; Obendorf, 2009; Harris, 2017; Colborne, 2017). In this case, we argue that strengthening contextual bias can be a remedy to mitigate the problem because it can naturally autofill some visual components that have not been explicitly conditioned, as shown in Fig. 2 (a).

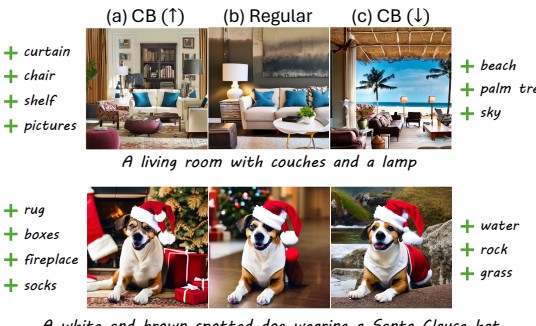

Figure 2: Visualizations of the effects of controlling contextual bias.

Second, given a scene description, if non-trivial and diverse object combinations are desired from the generated images, we claim that weakening contextual bias can be a solution. This is because it can smooth out the learned correlation, extrapolating the object combinations of the generated sample, as shown in Fig. 2 (c). In practice, this can be useful in creative image generation (Zylinska, 2020) and ideation (Paananen et al., 2023). It can also be useful in data augmentation (Shorten & Khoshgoftaar, 2019) since class imbalance (i.e., long-tailed problems) (Tang et al., 2020a) can be mitigated by augmenting class-balanced datasets which can be obtained by weakening contextual bias.

Dealing with contextual bias has been a widely-explored topic in causally-inspired literature (Deng & Zhang, 2022; Liu et al., 2022; Zhang et al., 2020; Yue et al., 2020; Tang et al., 2020b; Wang et al., 2020; Yang et al., 2023). Briefly, contextual bias is considered a confounder in a proposed causal graph, and the confounding effects can be mitigated by directly modeling interventional distribution or total direct effects with neural networks.

However, naively applying the existing causally-inspired methods to diffusion models is not practical enough due to their large-scale nature. To be specific, obtaining billions of data points and analyzing

---

[1]Per image, MSCOCO (Lin et al., 2014) has 5 captions and VisualGenome (Krishna et al., 2017) has an average of 42 region descriptions.

confounders, i.e., contextual bias, over massive-scale data can be arduous. Furthermore, large-scale diffusion models need to be redesigned to model the interventional distribution and need to be trained from scratch as well, which is far beyond reach in practice.

To this end, we propose two causally-motivated diffusion sampling frameworks that can strengthen and weaken the confounding effects, respectively, without finetuning or access to large-scale training data. Both approaches are inspired by the observations that pretrained large-scale models implicitly learned contextual bias, and thus the confounding variable and confounding effects can be retrieved by sampling from these models.

Briefly, we first design a causal graph representing the image generation process when training diffusion models (Fig. 3 (b)), where contextual bias is not explicitly modeled during training but is an unobserved confounder between the input text and the image. To strengthen the contextual bias in pretrained Diffusion Models, we leverage implicitly learned contextual bias of Large Language Models (LLM, e.g., Gemini (Gemini-Team et al., 2023)) to aid in retrieving and enhancing contextual bias (Fig. 3 (a)). To weaken the contextual bias, on the other hand, we first derive the interventional distribution and then approximate it by sampling from pretrained diffusion models and Vision Language Models (VLM, e.g., LLaVA (Liu et al., 2023; 2024b;a)). Following the derived sampling chain, we retrieve samples from the hidden confounder that pretrained diffusion models have learned implicitly. Next, we remove the confounding effects by adjusting for the retrieved confounder by backdoor adjustment formula (Pearl, 2009) (Fig. 3 (c)).

In summary, our contributions are as follows:

- We explicitly model contextual bias as a latent confounder and analyze how to retrieve this confounder using only pretrained models.
- We develop methods to increase and decrease the contextual bias using this latent confounder via marginalization and intervention respectively.
- We demonstrate quantitatively and qualitatively that our methods can improve the diversity or fidelity of the images while maintaining the content of the user-specified prompt. Furthermore, we showcase that these ideas can be applied alongside other controllability methods.
- We will release a dataset of 1,130,195 confounders (specifically co-occurring objects) that could be used in future contextual bias research.

## 2 BACKGROUND

**Diffusion models.** Denoising Diffusion Probabilistic Models (Sohl-Dickstein et al., 2015; Ho et al., 2020) (DDPM) are a class of probabilistic generative models mapping known distribution like Gaussian $X_T$ to unknown real distribution $X_0$. The reverse process is defined as a Markov Chain $p_\theta(X|Y=y) = \int p_\theta(x_{0:T}|y)dx_{1:T}$, where $p_\theta(x_{0:T}|y) = p(x_T)\prod_{t=1}^{T} p_\theta(x_{t-1}|x_t, y)$. Thus, to obtain the generated sample $x$ from the reverse process $p_\theta(X|y)$, we first need to sample from the standard Gaussian $p(x_T)$ and pass the sample through the reverse denoising steps from $t = T$ to $t = 1$. DDIM (Song et al., 2021) proposed a non-Markovian diffusion process, with which we can reduce the sampling time as well as deterministically sample from diffusion models. We use deterministic DDIM for sampling in all of the experiments. Latent Diffusion Models (Rombach et al., 2022) (LDM) introduces pretrained autoencoder (Esser et al., 2021) to diffusion models and use them as encoding/decoding modules to speed up trainig/sampling diffusion models.

**Contextual bias.** Though terminologies are not unified, the concept of contextual bias (i.e., the object co-occurrence statistics) has been a widely discussed topic in discriminative tasks such as visual question answering (Hendricks et al., 2018; Manjunatha et al., 2019; Cadene et al., 2019; Yang et al., 2021b), multi-label classification (Liu et al., 2022), scene graph generation (Tang et al., 2020b), few-shot learning (Yue et al., 2020), semantic segmentation (Zhang et al., 2020), and long-tailed classification (Tang et al., 2020a), where causal inference (Pearl, 2009) is one of the common solutions. On the other hand, in generative tasks, relatively less attention has been paid to the concept of contextual bias. It is used for increasing the interpretability of the classification result (Goyal et al., 2019; Lang et al., 2021) or augmenting data (Mao et al., 2021).

**Causal Inference in Generative Models.** Generative models typically aim to model observational data distribution, which has been widely explored showing remarkable performances. However, drawing samples beyond the observed data is fundamentally limited, where causal inference comes

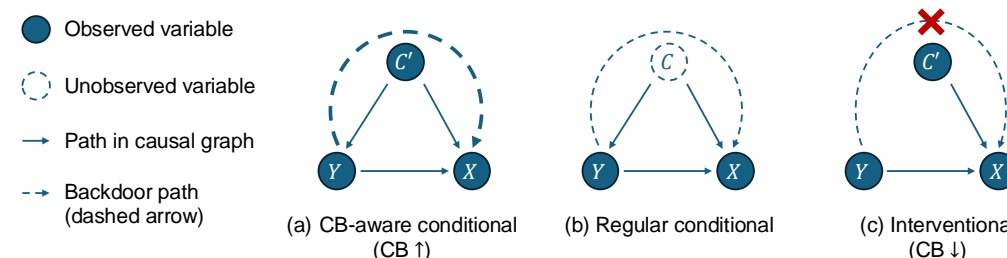

Figure 3: Illustrations of proposed causal graphs. Details are described in Sec. 3.1

into play. There have been many previous studies trying to combine generative models and Causal Inference; GANs (Kocaoglu et al., 2018; Shen et al.), VAE (Yang et al., 2021a; Karimi et al., 2020; Brehmer et al., 2022), normalizing flow (Pawlowski et al., 2020), autoregressive models (Khemakhem et al., 2021), and diffusion models (Chao et al., 2023; Sanchez & Tsaftaris, 2022; Varici et al., 2023; Lorch et al., 2024). Details are provided in Sec. C of Appendix.

As opposed to the related works, we aim to handle contextual bias without any training. By doing so, our results can have state-of-the-art performance of pretrained diffusion models while causal perspective can play a role in enhancing the controllability and interpretability of diffusion sampling.

**Heuristic approaches to handle contextual bias of diffusion models.** Bansal et al. (2022) proposed to add ethical interventions to the original prompt for diversity of the generated images. Zhang et al. (2023a) proposed a framework to optimize inclusive tokens to generate debiased images with equally-distributes attributes. Differently, our approach can be applied to any scenes and objects. Their methods are not scalable to arbitrary scenes/objects as their methods requires manually predefined ethical interventions (Bansal et al., 2022) or includes optimizing/finetuning a prompt embedding for each object/attribute (Zhang et al., 2023a).

More importantly, none of these explicitly define contextual bias in a causal framework. Our methods combine the principled nature of causal approaches, where assumptions are explicit and the proposed causal graphs are easily applicable to arbitrary scenes/objects and are scalable to more complex situations as shown in Fig. 6 and Fig. 8 (a-b) in Appendix.

## 3 APPROACH

In Sec. 3.1, we elaborate the proposed causal graphs illustrated in Fig. 3. We next describe the causally-motivated diffusion sampling methods in Sec. 3.2 and Sec. 3.3.

### 3.1 CAUSAL GRAPH

**Nodes.** Variables $Y$ and $X$ denote text prompt and image. $C$ is an unobserved confounder in training data. We assume that $C$ is an inaccessible hidden confounder since pretrained diffusion Models are already trained over billions of data. $C'$ is a retrieved confounder on a sampling basis. Our proposed methods aim to control (i.e., strengthen or weaken) the retrieved confounder following the techniques in Causal Inference (Pearl, 2009). Detailed methods for obtaining $C'$ are in Sec. 3.2 and Sec. 3.3.

**Edges $C \to (X, Y)$ and $C' \to (X, Y)$.** The edge $C \to (X, Y)$ represents the generation process of training data. As discussed in Sec. 1, object co-occurrence prevails in the real world, and thus we can easily come up with some cases of co-occurring objects, such as an oven, a stove, and a refrigerator. This indicates that contextual bias $C$ might have affected the formation of training data. Since $C$ is inaccessible though, we use the retrieved variable $C'$ as a confounder and control it.

**Edges $Y \to X$ and $(Y, C') \to X$.** The edge $Y \to X$ is a standard conditional diffusion sampling that does not consider any confounding effects (Fig. 3 (b)). On the other hand, the edge $(Y, C') \to X$ explicitly takes the retrieved confounding variable $C'$ as an additional condition to strengthen or weaken the confounding effects (Fig. 3 (a), (c)).

**Backdoor paths $(Y \leftarrow C' \to X)$ and $(Y \leftarrow C \to X)$.** In Fig. 3 (a) and (b), we can see that $Y$ and $X$ are not d-separated because there is a backdoor path from $Y$ to $X$ through $C'$ or $C$. In other words, a generated image $x$ can be caused by $y$ (as we expect), but also caused by non-causal

association (i.e., the backdoor path through confounder). In Fig. 3 (c), on the other hand, we derive interventional distribution to cut off the backdoor path, which indicates we can expect that the confounding effects can be mitigated.

## 3.2 CB-AWARE CONDITIONAL

The regular diffusion sampling process can be represented as $p(X|Y = y)$ which does not consider any confounding variables in the generation process. To strengthen the confounding effects as shown in Fig. 3 (a), we first need to retrieve a sample of the confounding variable $C'$ and condition the sample $c'$ on pretrained diffusion models. Formally, this can be formulated as:

$$p(X|Y = y) = \sum_c p(X|Y = y, C = c)p(C = c|Y = y) \tag{1}$$

$$\approx \sum_{c'} p_\theta(X|Y = y, C' = c') \underbrace{p_\psi(C' = c'|Y = y)}_{\text{Observational contextual bias}} \tag{2}$$

$$= \mathbb{E}_{c'|y}\left[p_\theta(X|Y = y, C' = c')\right], \tag{3}$$

where $p_\psi(C' = c'|Y = y)$ is an observational contextual bias, implemented as an LLM (e.g., Gemini (Gemini-Team et al., 2023)). Note that the variable $C$ in Eq. 1 is intractable, and thus we use the retrieved confounding variable $C'$ in other equations. Eq. 3 can be seen as a mixture distribution, and thus we can sample an image $x$ from the mixture by conditioning $c' \sim p_\psi(C' = c'|Y = y)$ and the prompt $y$ on pretrained diffusion models $p_\theta(\cdot)$. Since $c'$ is a set of co-occurring objects in text format, it can be conditioned together with $y$ in the sampling process. To obtain $c'$, specifically, we used a query of f`What objects can be put in the scene of "{y}"? Answer {k} objects in English in one line with comma.` to obtain potential co-occurring objects $c'$ for the scene of $y$. An example of a sampled $c'$ is [`couch`, `rug`, `bookshelf`, ... , `fireplace`] given a $y$ of `A living room filled with furniture and a fire place`. We empirically use $k = 10$.

## 3.3 INTERVENTIONAL

To weaken the contextual bias as shown in Fig. 3 (c), we need to cut off the confounding effects along the backdoor path $Y \leftarrow C' \rightarrow X$ by intervening on $Y$. To be concrete, we first apply the *do*-operator (Pearl, 2009) to get the interventional. We next identify the causal effect of $Y$ on $X$ by adjusting for the confounder $C'$ following the backdoor criterion (Hernán & Robins, 2010; Pearl, 2009). Formally, it can be formulated as:

$$p(X|\text{do}(Y = y)) = \sum_c p(X|Y = y, C = c) \underbrace{p(C = c)}_{\text{CB Prior}}, \tag{4}$$

where the contextual-bias (CB) prior $p(C = c)$ is intractable. Hence, we derive a method for obtaining a CB prior on a sampling basis (detailed derivations are provided in Sec. A):

$$p(C = c) = \sum_{y'} \sum_{x''} p(C = c|Y = y', X = x'')p(X = x''|Y = y')$$
$$\sum_{x'} p(Y = y'|X = x')p(X = x'). \tag{5}$$

By leveraging pretrained models to approximate the sampling chains, we can obtain a sample of the retrieved confounder $C'$:

$$p(C' = c') \approx \sum_{y'} \sum_{x''} p_\phi(C' = c'|Y = y', X = x'')p_\theta(X = x''|Y = y')$$
$$\sum_{x'} p_\phi(Y = y'|X = x')p_\theta(X = x'), \tag{6}$$

where $p_\theta(\cdot)$ is the reverse process of pretrained diffusion models $\epsilon_\theta$, and $p_\phi(\cdot)$ is pretrained Vision Language Models (VLM), such as LLaVA (Liu et al., 2023; 2024b;a). To be concrete, from the rightmost term to the left, $x'$ indicates an unconditionally generated image. We next leverage LLaVA to retrieve the pretrained knowledge of diffusion models in text form $y'$. A query of `Shortly describe the scene in one sentence` is used. We then obtain conditionally generated image $x''$ (guided by $y'$), with which we can finally obtain a sample from $p(C' = c')$ by using LLaVA.

A query of `What objects are in the image? Answer in one line with a comma.` is used. Empirically, we use $p_\phi(C' = c'|X = x'')$ instead of $p_\phi(C' = c'|Y = y', X = x'')$ since the information in $y'$ and $x''$ are almost identical.

By replacing the CB prior $p(C = c)$ with the retrieved CB prior $p(C' = c')$, Eq. 4 is reformulated as:

$$p(X|\text{do}(Y = y)) \approx \sum_{c'} p_\theta(X|Y = y, C' = c') \underbrace{p(C' = c')}_{\text{Retrieved CB Prior}} = \mathbb{E}_{c'}\left[p_\theta(X|Y = y, C' = c')\right].$$

(7)

To sample $x$ from the interventional in the above equation, we first sample $c'$ from the mixture distribution in Eq. 6. After sampling $c'$, we can get an image $x$ by conditioning $c'$ and $y$ to pretrained diffusion models, similar to Sec. 3.2.

**Speeding up the sampling $c'$.** As the readers might notice, the sampling chains in Eq. 7 and Eq. 6 can be slow and computationally expensive. Thus, we preprocess the sampling chains over 1,130,195 samples and present the empirical distribution $p(C')$ to boost the speed of the sampling process. Naively sampling by going through all the sampling chains takes 12 seconds with a single RTX A5000 while sampling with the precomputed $p(C')$ takes less than 1 second which is doable considering that sampling from diffusion models itself takes 3-4 seconds.

## 4 EXPERIMENTS

In this section, we first qualitatively evaluate the results. We next answer four important questions to quantitatively evaluate our proposed methods compared to the regular diffusion sampling. Experiment settings including the dataset and the measure are provided in Appendix Sec. B.

**Qualitatively exploring the effects of proposed two methods.** Fig. 4 shows the comparison results of our proposed methods and regular diffusion sampling. VG results are for simulating only a little piece of information is given. For example, (n = 1) indicates a case in which only a single object is given. COCO results are for simulating a natural querying by using a caption. The 'Real' column shows a paired real image with the prompt, and the 'Reg' indicates the regular diffusion sampling without considering any contextual bias. Overall, we can see that strengthening contextual bias (CB+) tends to make a natural scene by adding some objects that can be naturally put together. For example, given 'bedroom', 'bed', and 'wall' in the second row, the result from CB+ contains many objects related to the query including windows, curtains, and a vanity. On the other hand, the regular diffusion sampling contains fewer objects related to the query and does not include some that could have naturally filled the scene.

As for weakening contextual bias (CB-), it cuts off the learned contextual bias of pretrained diffusion models by intervening on the conditioning variable $Y = y$, and adding contextual bias which is disconnected from the condition $y$. For example, the result in the center column of VG (n = 7) in Fig. 4 has motorcycles given some words related to kitchen. The other example is in the center column of COCO which shows an outdoor toilet in front of the tree. These examples show that CB- is effective in extrapolating co-occurring object combinations beyond data-driven correlation. In the same context, CB- also can be useful in inspiring our creativity as shown in the boat in the street filled with water in the center column of VG (n = 5), and the subway tunnel covered by moss in the left column of VG (n = 7).

Some questions we can have here are: 1. How does controlling contextual bias affect the generation performance? 2. Can CB+ autofill the scene? 3. Does the generated image maintain well the original prompt information? 4. How much does it diverge from the regular sampling?

**Are the results from CB+ and CB- realistic enough?** We first measure FID to know how generation quality is affected by controlling contextual bias. For each CB+, Reg, and CB-, we generate 10k samples from the captions of COCO dataset. As can be seen in Tab. 1, weakening contextual bias (CB-) consistently shows better performance in FID. Generally speaking, FID incorporates both quality and diversity

Table 1: FID comparisons. VG (1) indicates VG dataset with one object (n=1).

|  | FID ($\downarrow$) | | |
| --- | --- | --- | --- |
|  | CB+ | Reg | CB- |
| VG (1) | 68.62 | 74.44 | **52.92** |
| VG (3) | 54.54 | 57.67 | **48.42** |
| VG (5) | 50.49 | 47.65 | **46.08** |
| VG (7) | 46.38 | 43.92 | **43.45** |
| COCO | 27.08 | 26.25 | **25.79** |

into one measure. Thus, increasing diversity appropriately can improve FID. We believe the better performance of CB- is because it can diversify the generated objects by leveraging the retrieved confounder $c' \sim p(c')$. In fundamental, $c'$ in the mixture distribution $\mathbb{E}_{c'}[p_\theta(X|Y = y, C' = c')]$ can provide independent information of $y$, and thus CB- can systematically generate more diverse results than Reg. The increased diversity of CB- can be also verified by the higher LPIPS score in Tab. 4.

Interestingly, strengthening contextual bias (CB+) also shows better performance in FID if a small amount of information is provided as shown in VG (1-3) in Tab. 1 (marked in blue). We think this is because FID takes into account both the mean and variance of generated data, and CB+ in the low information cases might help the generated samples match the mean better even if it reduces the covariance/diversity to some extent.

**Can CB+ actually autofill the scene?** To further verify the benefit of CB+ over Reg, we also measure CLIP scores in VG settings. CLIP score is used to measure the effectiveness of CB+ in recovering (i.e., autofilling) the complex scene given a little information. Specifically, we first preprocess the caption labels of VG dataset by SpaCy and use the extracted noun tokens per image as ground-truth objects. Next, we generate 1,000 samples for each

Table 2: Scene recovery performance.

|  | CLIP (Text Sim) | | CLIP (Image) | |
|---|---|---|---|---|
|  | CB+ | Reg | CB+ | Reg |
| VG (1) | **0.62** | 0.61 | **21.23** | 20.41 |
| VG (2) | **0.63** | 0.62 | **22.66** | 22.61 |
| VG (3) | 0.63 | **0.64** | 23.11 | **23.74** |
| VG (5) | 0.64 | **0.65** | 23.92 | **24.49** |

setting (CB+, Reg) and use LLaVA to get what objects are generated. We next use CLIP text encoder to compute the feature-level cosine similarity between the ground-truth and the generated objects. We also report CLIP scores between the generated images with ground-truth objects.

The results are shown in Tab. 2. CB+ shows better performance in recovering the original scene only given a little information (e.g., VG 1-2). Indeed, this matches our intuition because adding contextual bias can be useful if a prompt is not specific enough to describe the whole scene. If it is concrete enough, the autofilled contextual bias does not play a crucial role and rather can degenerate the CLIP performance. This is because the autofilled objects can be diverged by adding unnecessary contextual bias, e.g., tree and bench in VG (5) in Fig. 4.

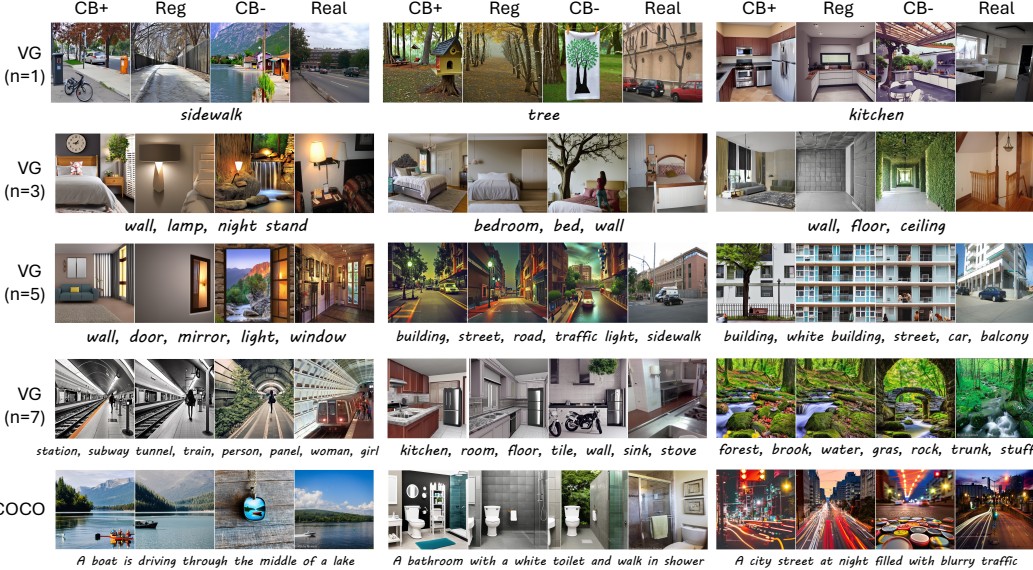

Figure 4: Qualitative comparisons of our proposed methods with the diffusion regular sampling.

**Can original $y$ be preserved?** As shown in Fig. 3, the retrieved confounder $c'$ is explicitly modeled in our causal framework, and it is given as an additional condition to pretrained diffusion models. Even though it gives an additional dimension of controllability on contextual bias, this can dilute the information of the original $y$ in practice. To measure how much the original $y$ is preserved, we conduct an experiment comparing the word tokens in $y$ and the retrieved word tokens from the generated

Table 3: Quantitative comparisons on prompt preservation. Our proposed methods yield more realistic and diverse results in exchange for the insignificant loss of prompt preservation.

|  | CB+ | Reg | CB- |
|---|---|---|---|
| Acc (%) | 73.0 | 75.2 | 71.2 |

Table 4: Quantiative results on LPIPS and mutual information. $I_n$ indicates mutual information computed over n-gram based histograms.

|  | LPIPS | $I_1$ | $I_2$ |
|---|---|---|---|
| CB+ | 0.4853 | 3.83 | 3.24 |
| CB- | 0.5888 | 3.33 | 2.32 |

images. Specifically, we first generate 10,000 samples for each setting (CB+, Reg, CB-) with the COCO test+val dataset. Each sample for (CB+, Reg, CB-) is from the same prompt $y$ and the random noise $z_T$ but with a different $c'$; CB+ takes $c' \sim p(c'|y)$ as input, and CB- takes $c' \sim p(c')$, as shown in Eq. (3) and Eq. (7) in the main paper. Reg, on the other hand, takes nothing related to $c'$ as input, i.e., $p(X|Y = y, C' = \text{Null})$. We next leverage LLaVA to extract the object information $\hat{c}$ from the 10,000 generated images per setting by asking "What objects are in the image? Answer in one line with a comma." To measure the performance, we obtain $\tilde{c}$ from the prompt $y$ by using POS tagging "NOUN" from Spacy. For example, given a $y =$ "A man with a red helmet on a small moped on a dirt road.", we get $\tilde{c} =$ "{ man, helmet, dirt, road }". Finally, we measure the accuracy by comparing $\hat{c}$ and $\tilde{c}$. If one object is overlapped, it is counted as a correct sample. When comparing, we use SpaCy to $\hat{c}$ as well to put it in the same space with $\tilde{c}$.

Tab. 3 shows the comparison results. Not surprisingly, the regular sampling shows better performance than other methods because it is solely sampled from $y$. CB+ shows better performance than CB- but worse than the regular sampling in terms of preserving $y$. This is because the retrieved confounder $c'$ affects the generated result reducing the relative impact of $y$. For example, in the left column of VG ($n = 3$) in Fig. 4, the effects of conditions 'wall', 'lamp', 'night stand' become relatively smaller in CB+ than the regular result because the scene is filled with other contextual co-occurring objects.

CB- shows relatively the lowest accuracy, but the difference is not significant. In fundamental, the low accuracy can be understood by looking at the joint likelihood $p(c', y|x)$ that can be low because $c'$ and $y$ are less likely co-occur in the real world. Thus, the implicit classifier $p(c', y|x) \propto p(x|c', y)p(c', y)$, which is modeled as pretrained diffusion models, also can show worse performance, e.g., $y$ and $c'$ are combined unnaturally or either one of them can be ignored in the generation process. In the same context, even though the generated sample contains both $y$ and $c'$, a classifier can miss one of them because $p(c', y|x)$ can be a low-density region. An example would be the left column of COCO in Fig. 4. A classifier could miss 'a boat' and 'a lake' in the necklace. Another intuitive example could be the center column of VG ($n = 1$) where a towel is generated with a tree drawn on it.

**How much does the generated spectrum diverge?** We have observed that the generated samples by CB+ and CB- can be more diverse than the regular sampling method because we explicitly model the retrieved confounder $c'$ and apply it to the sampling process as an additional condition. To explore the phenomenon visually, we generate 10 images while fixing the starting point $x_T$ and the prompt $y$ but varying $c'$ following Eq. 2 and Eq. 7. The results are shown in Fig. 5. We observe that CB+ consistently fills the scene with related objects to the prompt. However, sometimes the diversity is limited especially when the output of the regular sampling already contains enough contextual bias of the scene, as shown in the bottommost row. On the other hand, CB- shows diverse scene compositions with various objects beyond contextual bias. A bookshelf on the snow-covered ground (third row) and an elephant behind the living room through the window (fifth row) are examples.

To quantify the divergence, we first measure the LPIPS of CB+ and CB-. To be specific, we generate 10k images by sampling 10 images per caption from COCO. The first thousand captions are used. LPIPS is measured by computing the averaged pair-wise feature distance between 45 pairs per caption (from $\binom{10}{2}$). Since we want to measure the generated spectrum obtained solely by $c'$, we fix other variables, such as $x_T$ and $y$, but only vary $c'$ (with a deterministic sampling of DDIM). Thus, LPIPS can be measured only for CB+ and CB- where $c'$ is sampled. The first column in Tab. 4 shows the results. As we expected, CB- shows more diverse outputs. This is because $c' \sim p(c')$ in Eq. 7 has a bigger variance than $c' \sim p(c'|y)$ in Eq. 2.

We next attempt to quantify how much the generated object distribution from our methods diverges from that of the regular sampling. To be concrete, by using the 10k images above (used in LPIPS), we make a thousand count histograms (e.g., over 10 samples per caption) for each of CB+, CB-, and Reg. We also leverage the notion of n-gram to make the histogram smoother and see the effects of a more

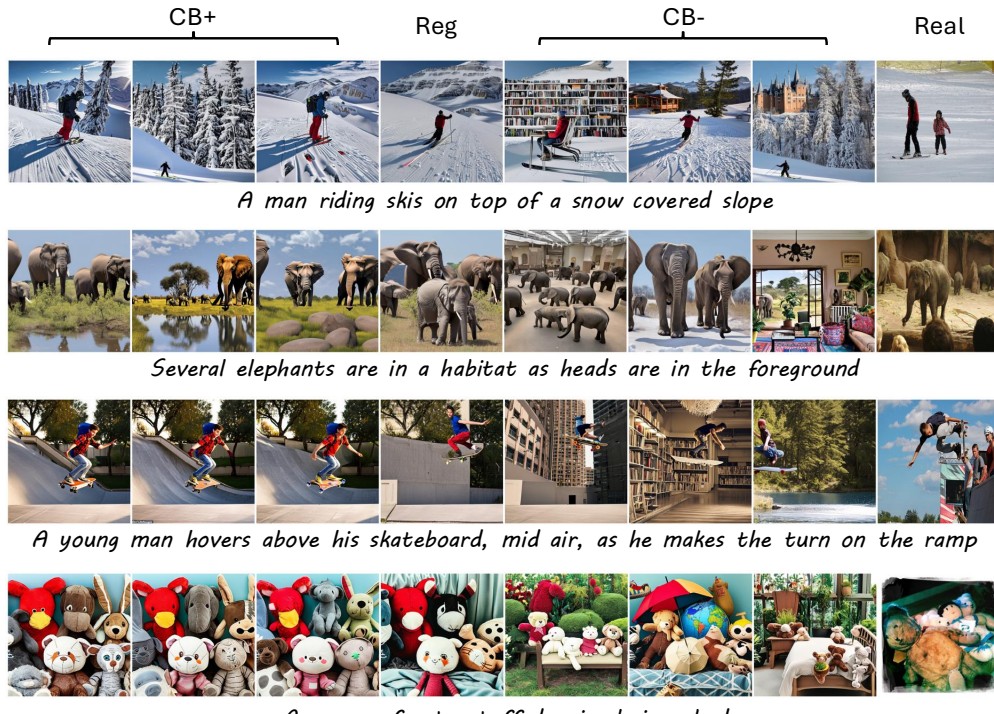

Figure 5: Qualitative comparisons for diverse image generation.

frequently concurring set of objects. For example, if a given data is [('a','b'),('a','b'),('b','c'),('c')], the count histogram before normalization would be {'a':2, 'b':3, 'c':2} in 1-gram. In 2-gram, however, the count histogram becomes {('a','b'):2,('b','c'):1}, which can shrink the distribution and show better a set of co-occurring objects. Finally, we measure mutual information (I) between the histograms of our methods and the regular one, i.e., I(CB+;Reg) and I(CB-;Reg). The results are shown in Tab. 4. The lower the mutual information I(A;B) is, the more independent the object histograms A and B are. We can see that CB- consistently shows lower mutual information than CB+. This means that the objects generated from CB- are more independent of the objects from the regular sampling.

We can also observe that the gap between CB+ and CB-in 1-gram ($I_1$) is increased in 2-gram ($I_2$) from 0.5 to 0.92. This indicates that the objects from CB- become even more independent of those from the regular sampling if we only consider more frequently co-occurring objects. In other words, even though both CB+ and CB- leverage the retrieved confounder $c'$ as an additional condition, the $c'$ from CB- guides the generation process to have more prompt-unrelated objects, verifying the effects of our interventional sampling method.

**Applcations.** Our proposed sampling methods are general, hence, we further apply our proposed methods to other diffusion methods for controllability (ControlNet (Zhang et al., 2023b) and DEADiff (Qi et al., 2024)). The results are shown in Fig. 6. Briefly, ControlNet (Zhang et al., 2023b) proposed a method for fine-tuning additional conditioning encoders on pretrained diffusion models. We use their pretrained depth ControlNet to give content information as a condition. DEADiff proposed a framework to disentangle the style and the content representations on top of pretrained diffusion models. We leverage their style representations to guide the sampling process. The results are provided in Fig. 6.

We observe that our proposed sampling methods are harmonized well with both ControlNet and DEADiff. For

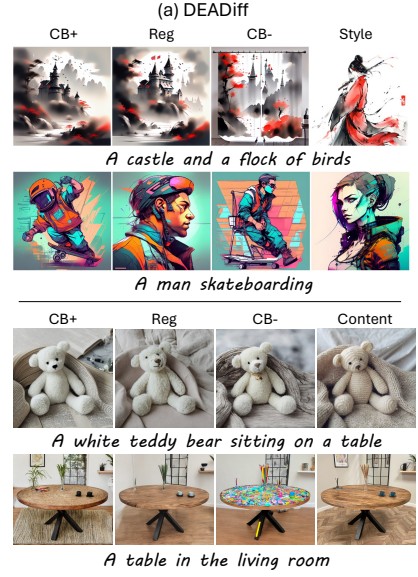

Figure 6: Our proposed sampling methods are general and thus can be easily adapted to other methods.

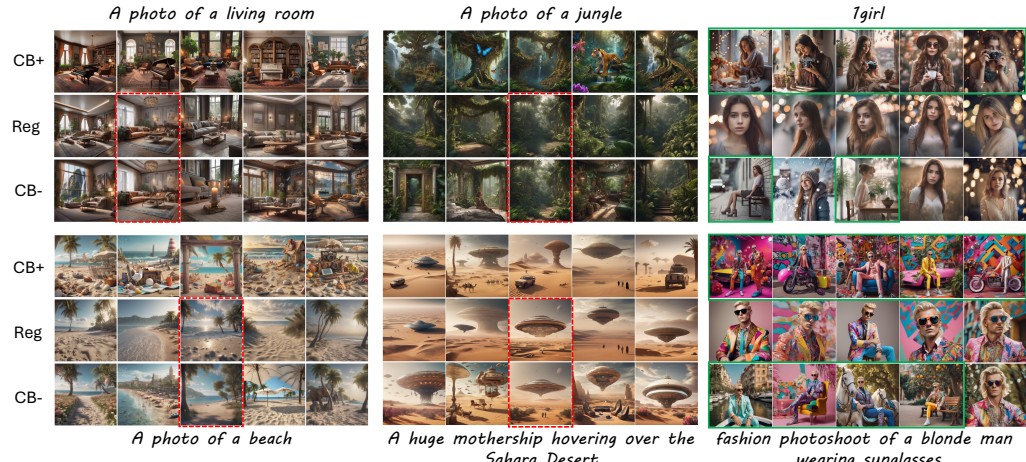

CB+
Reg
CB-

CB+
Reg
CB-

*A photo of a living room*  *A photo of a jungle*  *1girl*

*A photo of a beach*  *A huge mothership hovering over the Sahara Desert*  *fashion photoshoot of a blonde man wearing sunglasses*

Figure 7: Randomly sampled results from SDXL.

example, while the style from DEADiff is transferred successfully, in the second row, the CB+ result contains better contextual information than the regular sampling considering that the query is 'skateboarding'. CB- also shows an interesting result especially in the first row by drawing the castle in the curtain. With ControlNet, the generated table from CB- has colorful drawings and pencils while maintaining the original shape of the content. The results from CB+ also show good performance in adding contextual bias by generating a rug and a painting around the desk.

**SDXL results.** We further showcase the randomly generated samples from SDXL in Fig. 7. Detailed descriptions are provided in Appendix Sec. D.1.

## 5 LIMITATIONS & SOCIETAL IMPACTS & CONCLUSION

**Limitations.** Our proposed sampling methods have two limitations. As seen in Table 3 of the main paper, the explicitly modeled $c'$ from CB+ and CB- can sometimes yield slightly degraded prompt preservation which is expected when increasing or decreasing contextual bias. This can also be observed in the green box of Fig. 7. For example, $y = $ "1girl", but the results of CB+ and CB- contain other objects or contexts not in the $y$. Secondly, if the sampled $c'$ is semantically too far from $y$, the generated results sometimes ignore the added information from $c'$. The examples are shown in the dotted red boxes in Fig. 7. For instance, under "A photo of a living room", $c'$ for CB- in the red box is "horses, stadium, people". Additionally, our proposed frameworks are dependent on LLM or VLM which can introduce additional bias. However, We believe the bias from LLM and VLM can be closely related to the bias in pretrained diffusion models because all foundation-level models are mostly trained on "Internet data", which has its own set of biases. A more detailed discussion is in Sec. D.2 in Appendix.

**Societal Impacts.** Our method could have a potential societal impact as we can adjust the context of the generated images. However, we believe the developed technology would bring more good than harm by demonstrating that it is possible for such adjustment, which informs people to be more mindful of the content generated from AI models.

**Conclusion.** In this work, we proposed two sampling methods for controlling contextual bias in diffusion sampling frameworks based on Causal Inference. We define the contextual bias issue in the pretrained diffusion models. Since it can be a double-edged sword depending on the situation, we propose to control the contextual bias during the generation process. We first propose a causal graph where the contextual bias is explicitly modeled. We propose two sampling methods for retrieving contextual bias to strengthen/weaken the confounding effects. By involving the retrieved confounder in the generation process, we show that the generated results can be more diverse and realistic in exchange for the insignificant loss of prompt preservation. We also show that our proposed method is general and easily adaptable to other diffusion works based on pretrained diffusion models. We hope our work and novel perspective can inspire future research on causally motivated diffusion models.

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

## A   DERIVATIONS FOR RETRIEVING CONTEXTUAL PRIOR

$$p(C = c) = \sum_{y'} p(C = c, Y = y') \tag{8}$$

$$= \sum_{y'} p(C = c|Y = y')p(Y = y') \tag{9}$$

$$= \sum_{y'} p(C = c|Y = y') \sum_{x'} p(Y = y', X = x') \tag{10}$$

$$= \sum_{y'} p(C = c|Y = y') \sum_{x'} p(Y = y'|X = x')p(X = x') \tag{11}$$

$$= \sum_{y'} \sum_{x''} p(C = c, X = x''|Y = y') \sum_{x'} p(Y = y'|X = x')p(X = x') \tag{12}$$

$$= \sum_{y'} \sum_{x''} p(C = c|Y = y', X = x'')p(X = x''|Y = y') \sum_{x'} p(Y = y'|X = x')p(X = x')$$
$$\tag{13}$$

## B   EXPERIMENT SETTINGS

**Dataset.**   VisualGenome (Krishna et al., 2017) (VG) data has an average of 35 objects and their bounding boxes per image. To simulate a query with little information, we extract n objects depending on the bounding box area. For example, for sampling one object (e.g., VG (1)), we choose the biggest object assuming that it can represent the whole scene of the image better than other smaller objects. COCO (Lin et al., 2014) data has 5 captions per image among which we use the first caption to simulate a more natural querying.

**Measure.**   Fréchet inception distance (Heusel et al., 2017) (FID) is a widely used measure for evaluating the realism of the generated image. The lower FID score is, the closer the feature distribution of the generated images is to the real distribution, which means the generated images can be seen as more realistic. We use the whole validation set of COCO 2014 on the real side when measuring FID. 10k generated images are used for measuring FID. LPIPS (Zhang et al., 2018) is to measure the diversity of the generated images. To measure LPIPS, we generate 10 images per prompt and noise. For example, given a fixed prompt $y$ and a noise $x_T$, we sample 10 images (1000 captions in COCO) by sampling $c'$ from Eq. 2 and Eq. 7. We next compute the pair-wise LPIPS.

## C   RELATED WORKS

**Generative Models with Causal Inference.**

There have been many previous studies trying to combine Generative Models and Causal Inference. CausalGAN (Kocaoglu et al., 2018) proposes a GAN-based framework to train a causal implicit generative model with a predefined causal graph. They show that CausalGAN can model interventional distributions beyond simple conditionals. Shen et al. proposes DEAR, a GAN-based disentangled learning method in Causal Representation Learning. DEAR leverages a Structural Causal Model (SCM) as a trainable prior of bidirectional Generative Models to do causal controllable generation. CausalVAE (Yang et al., 2021a) proposes a VAE-based framework to have disentangled latent representations. A causal layer is newly introduced where independent exogenous variables are transformed into causal endogenous variables. Model identifiability is further analyzed. In Karimi et al. (2020), Conditional VAE is used to estimate the conditional average treatment effect (CATE) of an intervention given only limited causal information (i.e., a causal graph without true SCMs). Brehmer et al. (2022) shows that weak supervision is sufficient to identify causal representations and their SCMs. Briefly, a paired data before-and-after randomly-sampled unknown intervention is needed. VAE is used to model their proposed Latent causal models (LCMs). Pawlowski et al. (2020) proposes a framework for creating SCMs of which components are made of deep learning modules. Their deep SCMs are designed to satisfy three rungs of the causation ladder (association, intervention,

counterfactuals), and normalizing flows and variational inference are used for tractable counterfactual inference. Inspired by the fact that an ordering over variables is defined in both autoregressive flow and causality, Khemakhem et al. (2021) proposes to use autoregressive flow in causality tasks, such as causal discovery, interventional and counterfactual predictions.

As for Causality-based diffusion models, Chao et al. (2023) proposes diffusion-based causal model (DCM) approximating both interventions and counterfactuals, which can be trained with a causal graph and observational data. As opposed to ours where each node of SCM is an input/output variable of diffusion models, each node of the given causal graph in Chao et al. (2023) is modeled as diffusion models, i.e., a child node takes as input the output of the parent node. They further measure the accuracy of the counterfactual estimations and show that the estimations can be bounded with reasonable assumptions. Given observational data and a causal structure, Sanchez & Tsaftaris (2022) proposes a framework Diff-SCM for estimating causal effect. As opposed to ours where diffusion steps are not involved in SCMs, their framework suggests considering diffusion processes as Causal Models. During the inference, Diff-SCM can be used for sampling from the interventionals or estimating counterfactuals. They also present a metric for evaluating the estimated counterfactuals from their proposed framework. Varici et al. (2023) proposes an approach in Causal Representation Learning. Given an unknown linear transformation and indirectly observed causal latent variables, their aim is to recover the linear transformation (i.e., identifiable representation learning) and Directed Acyclic Graph (DAG) of the causal latent variables (i.e., causal structure learning). Inspired by the fact that the changes in the score function and the intervening effect are highly correlated, they propose a transformation-recovering method that detects minimal variations across different interventional environments given latent variables' score function. This work aims to have disentangled representations which is fundamentally different from our task aims (i.e., retrieving the hidden confounder, i.e., contextual bias, of pretrained diffusion models and increasing/decreasing the contextual bias under the Causal framework). Recently, Lorch et al. (2024) proposes an idea to replace the formalism of SCMs for causal modeling with stationary diffusion processes which are particular diffusion processes that admit a stable stationary distribution. The benefits of this framework allow cyclic causal dependencies and more flexible causal modeling. How to model interventions and a kernel-based approach for parameter estimation are proposed. This work differs from the methods and goals of our work since it aims to replace the SCM approach to causal modeling, while we aim to understand the image generation process under SCM to apply the knowledge in Causal Inference (i.e., defining a problem; explicitly modeling a contextual bias in SCMs, and resolving the problem; controlling the contextual bias in the generation process under the SCM). Additionally, it requires stationary diffusion models and does not consider image-based or pretrained diffusion models.

## D  ADDITIONAL RESULTS

### D.1  DESCRIPTIONS ON RANDOMLY SAMPLED SDXL RESULTS.

CB+ has a strong tendency to autofill the scene of $y$ with frequently co-occurring objects, i.e., $c' \sim p(c'|y)$. For example, from the first row of "A photo of a beach", we can see that a lot of beach-related objects (e.g., a camera, seashells, a beach umbrella) are generated together. In the case of CB-, we can observe that non-trivial objects are generated in the scene of $y$ because a novel set of co-occurring objects (i.e., $c' \sim p(c')$) is injected into the image generation process. For instance, from the last row of "A huge mother ship . . .", we can see that the generated images contain non-trivial co-occurring objects for the Desert such as fishes, flowers, chairs, and a cat.

### D.2  BIAS FROM LLM, VLM, AND DIFFUSION MODELS

It has been a widely discussed topic in LLM and VLM research (Bender et al., 2021; Seth et al., 2023) that large-scale models have various kinds of bias inherited from the training data. In the same context, diffusion foundation models are also trained with billions of training data which can inject bias into the models. In specific, diffusion models leverage pretrained LLM or VLM for obtaining conditional representations during the training process. We believe the bias from LLM and VLM can significantly affect diffusion models. Furthermore, diffusion foundation models, e.g., StableDiffusion are pretrained with LAION dataset which use CLIP to filter the dataset. We believe the bias from CLIP can affect the training data of diffusion foundation Models, which also can affect the bias of

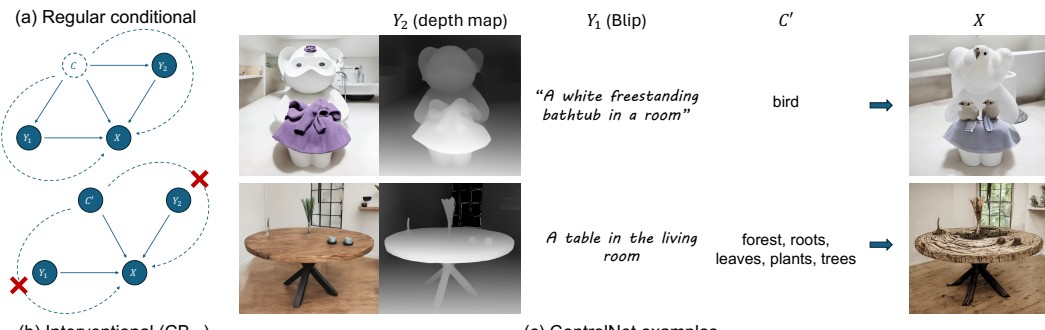

Figure 8: One of the benefits of Causal perspective in Diffuson sampling is that it is scalable to multiple conditions.

pretrained diffusion models. Thus, it is hard to see that the biases of diffusion models are not related to those of VLM and LLM.

Baseline 1: prompt engineering

"Autumn forest landscape, psychedelic style"

Ours

Figure 9: Additional baseline comparisons. Here, we randomly sample 20 samples per prompt and per setting and show all of them without cherry picking. The Baseline 1 is implemented by a simple prompt engineering. The query to Gemini is f"briefly modify the given prompt to be creative. Answer in one sentence.: '{prompt}' ". The engineered prompt is used as a text input.

Baseline 2: prompt engineering + negative prompt

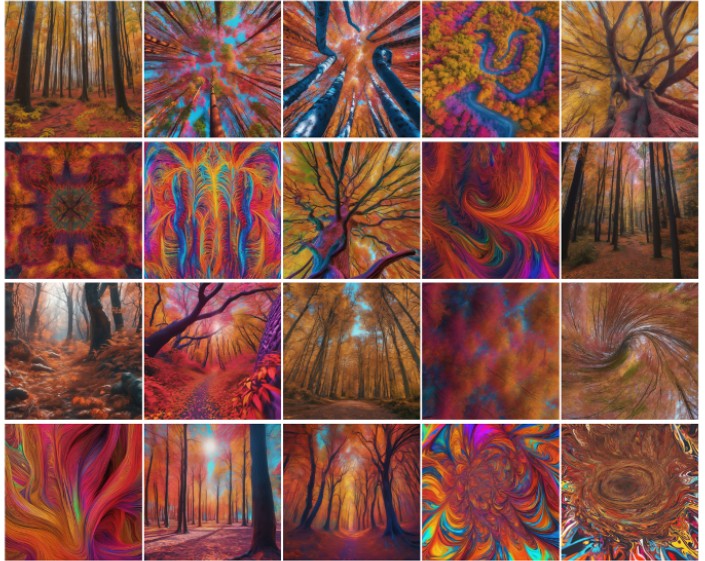

"Autumn forest landscape, psychedelic style"

Figure 10: Additional baseline comparisons. Here, we randomly sample 20 samples per prompt and per setting and show all of them without cherry picking. The Baseline 2 is implemented by prompt engineering and negative prompt techniques. The query to Gemini is f"What would be the frequently co-occurring objects that can be likely placed in the scene generated by the given prompt '{prompt}'? Do not answer the words mentioned in the prompt. Answer 10 objects except for {important_obj} one line with comma.", where important_obj is manually predefined as key object from the prompt. (e.g., "ancient dragon" from "A fantasy illustration of ancient dragon".) The answer is used as a negative prompt during the sampling process.

Figure 11: Additional baseline comparisons. Here, we randomly sample 20 samples per prompt and per setting and show all of them without cherry picking. The Baseline 1 is implemented by a simple prompt engineering. The query to Gemini is f"briefly modify the given prompt to be creative. Answer in one sentence.: '{prompt}' ". The engineered prompt is used as a text input.

Baseline 2: prompt engineering + negative prompt

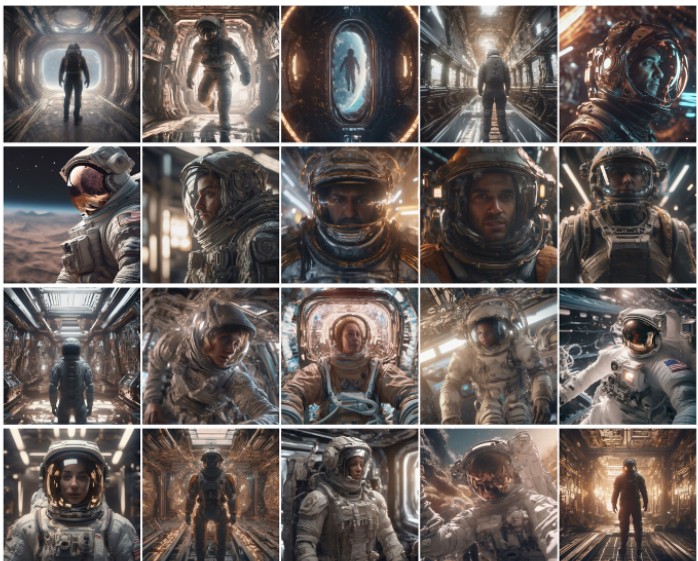

"photo of an astronaut"

Figure 12: Additional baseline comparisons. Here, we randomly sample 20 samples per prompt and per setting and show all of them without cherry picking. The Baseline 2 is implemented by prompt engineering and negative prompt techniques. The query to Gemini is f"What would be the frequently co-occurring objects that can be likely placed in the scene generated by the given prompt '{prompt}'? Do not answer the words mentioned in the prompt. Answer 10 objects except for {important_obj} one line with comma.", where important_obj is manually predefined as key object from the prompt (e.g., "ancient dragon" from "A fantasy illustration of ancient dragon".) The answer is used as a negative prompt during the sampling process.

Figure 13: Additional baseline comparisons. Here, we randomly sample 20 samples per prompt and per setting and show all of them without cherry picking. The Baseline 1 is implemented by a simple prompt engineering. The query to Gemini is f"briefly modify the given prompt to be creative. Answer in one sentence.: '{prompt}' ". The engineered prompt is used as a text input.

## Baseline 2: prompt engineering + negative prompt

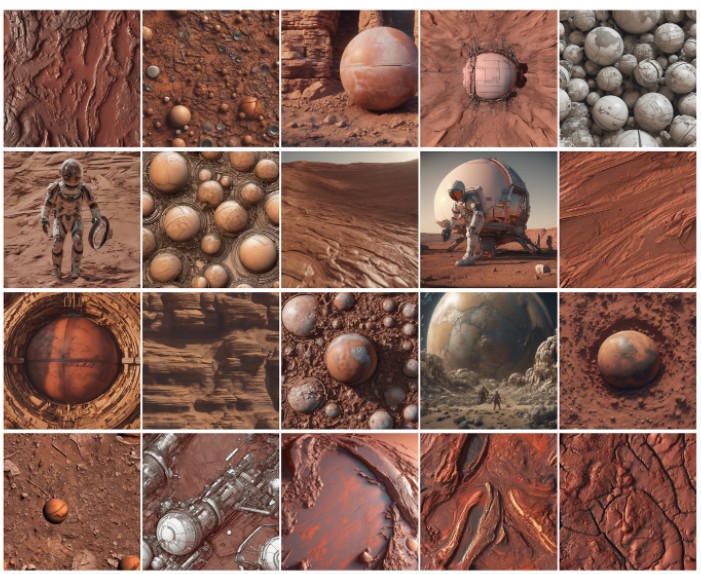

"photo of Mars"

Figure 14: Additional baseline comparisons. Here, we randomly sample 20 samples per prompt and per setting and show all of them without cherry picking. The Baseline 2 is implemented by prompt engineering and negative prompt techniques. The query to Gemini is f"What would be the frequently co-occurring objects that can be likely placed in the scene generated by the given prompt '{prompt}'? Do not answer the words mentioned in the prompt. Answer 10 objects except for {important_obj} one line with comma.", where important_obj is manually predefined as key object from the prompt. (e.g., "ancient dragon" from "A fantasy illustration of ancient dragon".) The answer is used as a negative prompt during the sampling process.

Figure 15: Additional baseline comparisons. Here, we randomly sample 20 samples per prompt and per setting and show all of them without cherry picking. The Baseline 1 is implemented by a simple prompt engineering. The query to Gemini is f`"briefly modify the given prompt to be creative. Answer in one sentence.: '{prompt}' "`. The engineered prompt is used as a text input.

Baseline 2: prompt engineering + negative prompt

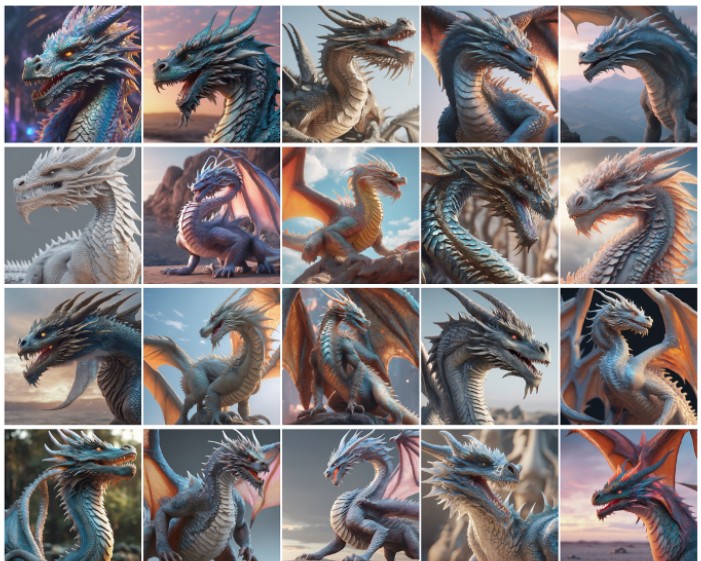

"A fantasy illustration of ancient dragon"

Figure 16: Additional baseline comparisons. Here, we randomly sample 20 samples per prompt and per setting and show all of them without cherry picking. The Baseline 2 is implemented by prompt engineering and negative prompt techniques. The query to Gemini is f"What would be the frequently co-occurring objects that can be likely placed in the scene generated by the given prompt '{prompt}'? Do not answer the words mentioned in the prompt. Answer 10 objects except for {important_obj} one line with comma.", where important_obj is manually predefined as key object from the prompt. (e.g., "ancient dragon" from "A fantasy illustration of ancient dragon".) The answer is used as a negative prompt during the sampling process.

## Baseline 1: prompt engineering

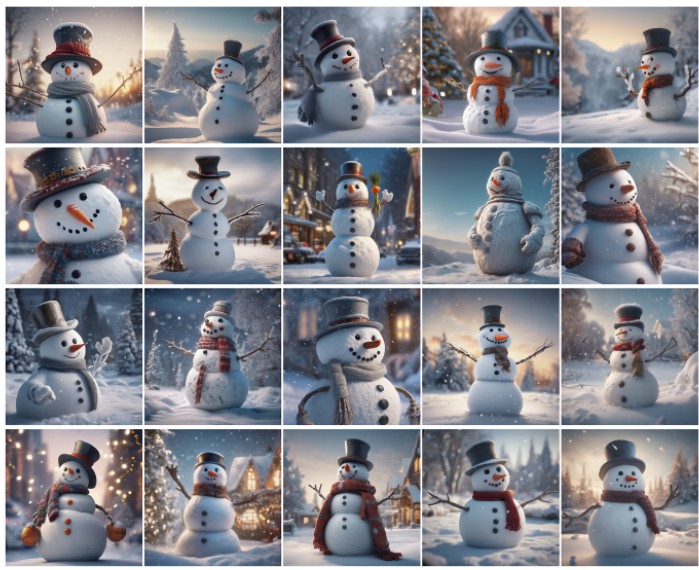

"photo of a snowman"

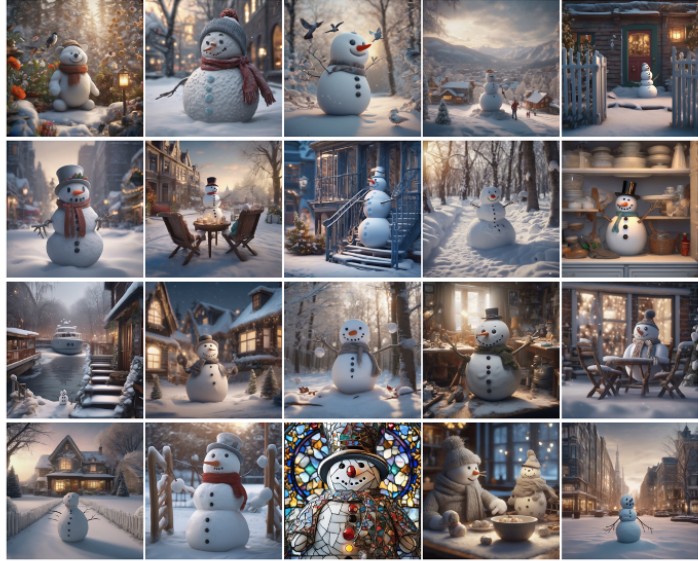

## Ours

Figure 17: Additional baseline comparisons. Here, we randomly sample 20 samples per prompt and per setting and show all of them without cherry picking. The Baseline 1 is implemented by a simple prompt engineering. The query to Gemini is f"briefly modify the given prompt to be creative. Answer in one sentence.: '{prompt}' ". The engineered prompt is used as a text input.

Baseline 2: prompt engineering + negative prompt

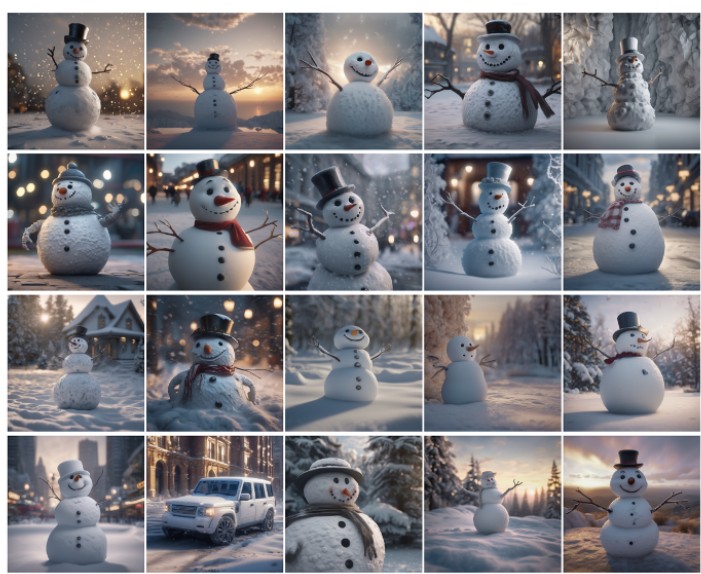

"photo of a snowman"

Figure 18: Additional baseline comparisons. Here, we randomly sample 20 samples per prompt and per setting and show all of them without cherry picking. The Baseline 2 is implemented by prompt engineering and negative prompt techniques. The query to Gemini is f"What would be the frequently co-occurring objects that can be likely placed in the scene generated by the given prompt '{prompt}'? Do not answer the words mentioned in the prompt. Answer 10 objects except for {important_obj} one line with comma.", where important_obj is manually predefined as key object from the prompt. (e.g., "ancient dragon" from "A fantasy illustration of ancient dragon".) The answer is used as a negative prompt during the sampling process.

Figure 19: Additional baseline comparisons. Here, we randomly sample 20 samples per prompt and per setting and show all of them without cherry picking. The Baseline 1 is implemented by a simple prompt engineering. The query to Gemini is f"briefly modify the given prompt to be creative. Answer in one sentence.: '{prompt}' ". The engineered prompt is used as a text input.

Baseline 2: prompt engineering + negative prompt

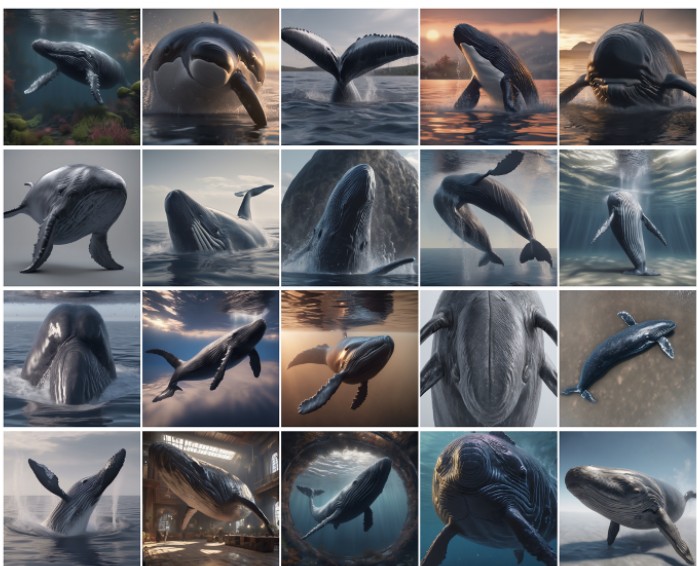

"photo of a whale"

Figure 20: Additional baseline comparisons. Here, we randomly sample 20 samples per prompt and per setting and show all of them without cherry picking. The Baseline 2 is implemented by prompt engineering and negative prompt techniques. The query to Gemini is f"What would be the frequently co-occurring objects that can be likely placed in the scene generated by the given prompt '{prompt}'? Do not answer the words mentioned in the prompt. Answer 10 objects except for {important_obj} one line with comma.", where important_obj is manually predefined as key object from the prompt. (e.g., "ancient dragon" from "A fantasy illustration of ancient dragon".) The answer is used as a negative prompt during the sampling process.

Figure 21: Additional baseline comparisons. Here, we randomly sample 20 samples per prompt and per setting and show all of them without cherry picking. The Baseline 1 is implemented by a simple prompt engineering. The query to Gemini is f"briefly modify the given prompt to be creative. Answer in one sentence.: '{prompt}' ". The engineered prompt is used as a text input.

Baseline 2: prompt engineering + negative prompt

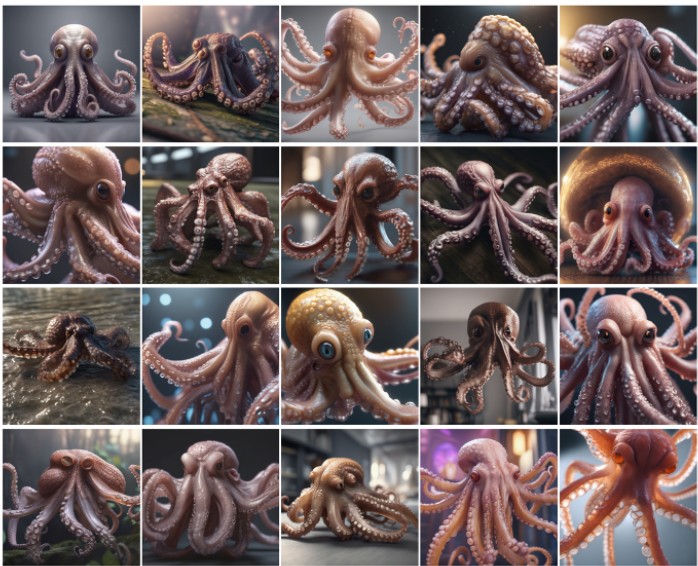

"photo of an octopus"

Figure 22: Additional baseline comparisons. Here, we randomly sample 20 samples per prompt and per setting and show all of them without cherry picking. The Baseline 2 is implemented by prompt engineering and negative prompt techniques. The query to Gemini is f"What would be the frequently co-occurring objects that can be likely placed in the scene generated by the given prompt '{prompt}'? Do not answer the words mentioned in the prompt. Answer 10 objects except for {important_obj} one line with comma.", where important_obj is manually predefined as key object from the prompt. (e.g., "ancient dragon" from "A fantasy illustration of ancient dragon".) The answer is used as a negative prompt during the sampling process.

Figure 23: Additional baseline comparisons. Here, we randomly sample 20 samples per prompt and per setting and show all of them without cherry picking. The Baseline 1 is implemented by a simple prompt engineering. The query to Gemini is f"briefly modify the given prompt to be creative. Answer in one sentence.: '{prompt}' ". The engineered prompt is used as a text input.

Baseline 2: prompt engineering + negative prompt

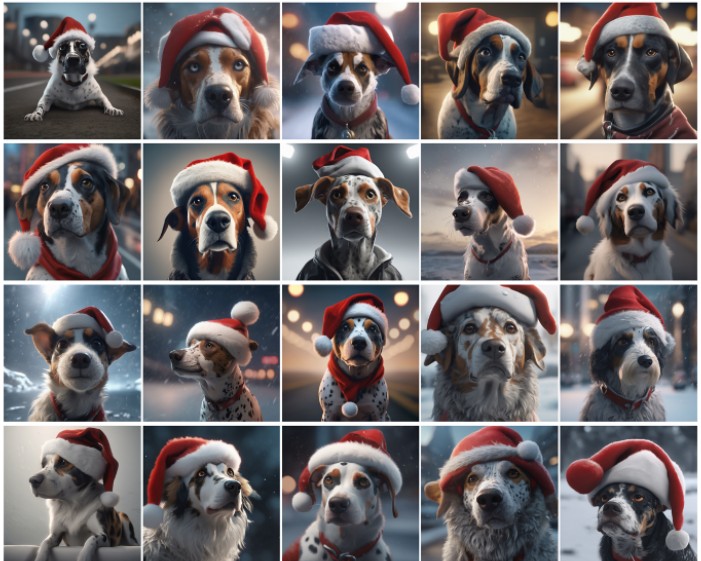

"A spotted dog wearing a Santa Claus hat"

Figure 24: Additional baseline comparisons. Here, we randomly sample 20 samples per prompt and per setting and show all of them without cherry picking. The Baseline 2 is implemented by prompt engineering and negative prompt techniques. The query to Gemini is f"What would be the frequently co-occurring objects that can be likely placed in the scene generated by the given prompt '{prompt}'? Do not answer the words mentioned in the prompt. Answer 10 objects except for {important_obj} one line with comma.", where important_obj is manually predefined as key object from the prompt. (e.g., "ancient dragon" from "A fantasy illustration of ancient dragon".) The answer is used as a negative prompt during the sampling process.

## Baseline 1: prompt engineering

"A cute rabbit"

## Ours

Figure 25: Additional baseline comparisons. Here, we randomly sample 20 samples per prompt and per setting and show all of them without cherry picking. The Baseline 1 is implemented by a simple prompt engineering. The query to Gemini is f"briefly modify the given prompt to be creative. Answer in one sentence.: '{prompt}' ". The engineered prompt is used as a text input.

Baseline 2: prompt engineering + negative prompt

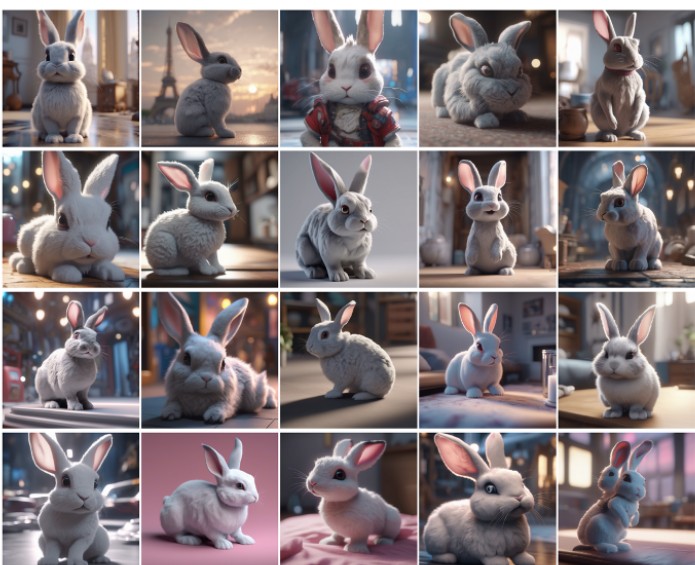

"A cute rabbit"

Figure 26: Additional baseline comparisons. Here, we randomly sample 20 samples per prompt and per setting and show all of them without cherry picking. The Baseline 2 is implemented by prompt engineering and negative prompt techniques. The query to Gemini is f"What would be the frequently co-occurring objects that can be likely placed in the scene generated by the given prompt '{prompt}'? Do not answer the words mentioned in the prompt. Answer 10 objects except for {important_obj} one line with comma.", where important_obj is manually predefined as key object from the prompt. (e.g., "ancient dragon" from "A fantasy illustration of ancient dragon".) The answer is used as a negative prompt during the sampling process.

