# OpenReview forum: "Causally Motivated Diffusion Sampling Frameworks for Harnessing Contextual Bias"
_ICLR.cc/2025/Conference — Submitted to ICLR 2025_

### Official Review · Reviewer_YF36 · 2024-10-25

**Soundness:** 3
**Presentation:** 2
**Contribution:** 2
**Rating:** 5
**Confidence:** 4

**Summary:**

This paper examines the phenomenon where diffusion models tend to generate images with certain preferences due to biases inherent in the training dataset. By applying causal learning framework, the influence of confounding factors is either enhanced or mitigated, thus making the generated images lean more toward "commonsense" or "counter-commonsense" representations. Notably, the authors leverage another form of bias, that present in LLMs/VLMs, to sample and estimate the distribution of confounding factors. And my understanding is that the core of this paper is not the diffusion model itself but rather the use of LLMs and VLMs to distinguish between commonsense and counter-commonsense content.

**Strengths:**

This work insightfully recognizes that contextual bias is not inherently negative. Contextual bias can be a crucial component in generating natural scenes, and enhancing control over it is essential. Through causal graph modeling, combined with the robust reasoning capabilities of LLMs/VLMs, they automatically (a key point) adjust the "amount of embedded commonsense" in the generated image results without any explicit training.

The highlight of this work is its approach to embedding causal intervention mechanisms in generative models, enabling automated confounder estimation to implement do-operations. While the use of causal graphs in the CV field has been widely discussed, the authors' combination of these methods with generative modeling is novel and yields promising results.

**Weaknesses:**

1. Hallucination of Large Model (Especially in Vision Language Model)

The limitations section highlights the constraints of VLMs. It is important to note that VLMs often experience hallucination issues. For example, when both a horse and a donkey appear in an image, the VLM may incorrectly label both as “donkey.” This is not a “beneficial” bias (as the authors point out in the paper) but rather a “harmful” hallucination issue, leading to inaccurate probability estimation when applying the Do operator.

Furthermore, how can we ensure that the commonsense knowledge (referred to as bias in the paper) embedded in LLMs/VLMs aligns with that in the Diffusion Model? After all, these models are trained on different datasets and strategy. If there is inconsistency in commonsense between them, then using LLMs/VLMs to estimate P(c) may not be an ideal approach.  For instance, if we aim to generate an image of a "bird," LLMs/VLMs might associate "bird" with "tree," while the Diffusion model may associate "bird" with "sky." In such cases, inconsistencies in bias arise. I believe expanding the discussion on this point could be beneficial to your work.

2.While the intervention mechanism is detailed, control over diffusion remains overly rough.

The core contribution of this paper is not in Diffusion itself but rather in using LLM/VLM combined with causal inference to extract a text segment "c" from the prompt. This prompt + "c" serves as a new conditional input for generation, with different "c" segments assigned distinct weights, ultimately leading to weighted summation on the latent space or score space.

However, this approach relies heavily on text prompt conditioning, limiting its effectiveness in handling complex cases. For instance, even with highly detailed prompts, the SD model may occasionally ignore specified objects in the prompt [1], which constrains the capability of the proposed algorithm.  A insightful work in this area can be found in [2], which delves deeper into the mechanisms of diffusion latent space, enabling the generation of counterintuitive or "counter-commonsense" images.

[1] Chefer, Hila, et al. “Attend-and-Excite: Attention-Based Semantic Guidance for Text-to-Image Diffusion Models.” ACM Transactions on Graphics, vol. 42, no. 4, July 2023, pp. 1–10. Crossref, https://doi.org/10.1145/3592116.

[2] Um, Soobin, and Jong Chul Ye. "Don't Play Favorites: Minority Guidance for Diffusion Models." arXiv preprint arXiv:2301.12334 (2023).

**Questions:**

1.To estimate the distribution of confounding factors, this method requires multiple sampling rounds from LLMs and VLMs; for example, generating a single image may involve at least 10 VLM calls, which is time-consuming.  In line 282, the author mentions pre-sampling methods.  Could you clarify the time required for this preprocessing, the approximate space complexity, and provide a detailed breakdown of the steps taken to reduce computation time? This part appears somewhat unclear.

2.Would it be beneficial to include a deeper discussion on the connection between diffusion bias and LLM/VLM bias, as this is a central focus of your research? Exploring whether a gap exists, if it can be quantified, and supporting this with further literature could enhance the work.

**Details Of Ethics Concerns:**

This paper does not have ethical concerns that are worrisome.

---

> ### Author Response · Authors · 2024-11-26
> **Response to Reviewer YF36**
>
> - (W2) **This approach relies heavily on text prompt conditioning, limiting its effectiveness in handling complex cases.** As the reviewer mentioned, text prompt based conditioning might have some limitations such as ignoring some objects in the complex prompt while the classifier-guidance-based approach [1] could bypass the issue. However, the approach of [1] is not scalable as they need to train a classifier per attribute (e.g., minority score) that can take noised image $x_t$ as input. They also require labeled dataset to train the classifier. On the other hand, ours does not require any additional training/finetuning process nor dataset preparation (assuming $p(C')$ is precomputed which is reasonable as we are going to release). Ours also can be applied to off-the-shelf pretrained Diffusion Models, which is a huge benefit in practice. Even though we might have the limitations of the long text conditioning, we believe our proposed methods have our own benefits compared to [1]. For example, our CB- can be easily applied to various scene generations as shown in Fig. 4, 5, and 7 in the main paper. However, it is very difficult for [1] to do the same thing as ours for two reasons:  1) it is unclear if their method can be used with pretrained StableDiffusion (SD), and 2) having a noise-aware classifier that can deal with thousands of diverse classes of SD is non-trivial.
>
> [1] Don't Play Favorites: Minority Guidance for Diffusion Models, ICLR'24
>
> - (Q1) **Clarifying the precomputing step of $p(C')$.** We apologize for the confusion. We provide a pseudo code for elaborating the process to retrieve (i.e., precompute) the contextual bias $p(C')$. Please see the Clarifications in "Response to all reviewers". As for the required time, for (a) unconditional image generation and (c) conditional image generation steps, we respectively use 5 days with 10 GPUs (RTX A5000). For (b) and (d) steps, we respectively use 2-3 days with 10 GPUs (RTX A5000). In total, we roughly used 15 days to obtain the final $p(C')$. As for the approximated space complexity, we use 1 TB for storing 1,130,195 images. (Note that one million images are obtained from the (a) step and can be removed after (b) step,  and another one million images are obtained from the (c) step.)
>
> - (W1, Q2) **Bias of Diffusion, VLM, and LLM** As the reviewer mentioned, our work assume that the bias of diffusion models are highly related to VLM and LLM. As mentioned in Section D.2 of the main paper, we tried to justify our assumptions by the fact that many large-scale models are heavily connected to one another because they share a public dataset, and LLMs could be used to generate text caption data. More specifically, diffusion models leverage pretrained LLM or VLM for obtaining text representations during the training process, from which we can suppose diffusion models are significantly affected by pretrained LLM or VLM. Furthermore, StableDiffusion and LLaVA are pretrained with LAION dataset which use CLIP to preprocess the data. That is, the bias from CLIP can affect the bias of pretrained StableDiffusion and LLaVA. Thus, while we agree that the contextual bias may not be exactly the same, they are related enough to be significantly useful as seen by our experiments. We are happy to include more discussion of this in the final paper.

---

> > ### Comment · Reviewer_YF36 · 2024-11-27
> >
> > Thanks the authors for a prompt reply! I appreciate your effort. However, I still have some unfixed concerns or questions.
> >
> > Regarding W2: control mechanism over diffusion remains overly rough.
> >
> > For instance, for a weak diffusion model, its understanding of the prompt is quite superficial. Even if your method makes additional modifications to the prompt (such as adding many counterfactual entities), it may not accurately reflect in the generated images, as discussed in works like "Attend and Excite." This limitation is due to the model's inherent capabilities and the sampling method, and cannot be solely improved by modifying the prompt. Therefore, I believe that focusing solely on the text prompt seems too crude an approach.
> >
> > Additionally, on this point, we share a similar view with reviewer pXtd's, as relying solely on manipulating the text prompt seems somewhat simplistic. It is essential for this work to demonstrate the necessity of this approach.
> >
> > ---
> >
> >
> > Regarding Q1: the space complexity of preprocessing.
> >
> > For this task, pre-storing 1TB of datasets seems excessively costly.
> >
> > ---

---

> ### Author Response · Authors · 2024-12-04
> **Second Response to Reviewer YF36 (1/2)**
>
> > control mechanism over diffusion remains overly rough. For instance, for a weak diffusion model, its understanding of the prompt is quite superficial. Even if your method makes additional modifications to the prompt, it may not accurately reflect in the generated images.
>
> As the reviewer pointed out, weak text-guided Diffusion Models (e.g., SD 1.x or 2.x) could have an issue in applying multiple objects (c.f., catastrophic neglect by [1]).
>
> However, as the reviewer might already know, there have been  some good research papers dealing with such issues [1-3]. Since their methods are implemented on top of the pretrained text-guided diffusion models as ours is, we can easily apply, let’s say, Attend-and-Excite during the sampling process to strengthen the subject tokens. For example, suppose a prompt $y$ is "A cat and a dog", and a sampled contextual bias $c’$ is "library". As shown in the rightmost column of Fig. 6 in Attend-and-Excite, we can see that the limitation (i.e., catastrophic neglect) of the text-based guidance could be mitigated successfully. Again, our method is not exclusive with the existing works [1-3], and thus we can mitigate the limitations of text-based conditioning by combining those works with our methods. Please note that the problems that the previous studies [1-3] aim to solve are far from the problem we are solving $^{1)}$, and thus combining [1-3] with our method does not affect our technical novelties.
>
> Secondly, the stronger version of Diffusion Models (e.g., SDXL) can deal with such issues better. Considering that the power of Diffusion Models will get stronger, we believe the weakness of text-based guidance will be even more mitigated at the end of the day.
>
> To summarize, the problem the reviewer pointed out (i.e., catastrophic neglect) can be mitigated by advanced sampling techniques [1-3] or by enhanced power of the base models (e.g., SDXL). Those techniques can be naturally combined with our proposed methods without degrading our technical novelties.
>
>
> [1] Attend-and-Excite: Attention-Based Semantic Guidance for Text-to-Image Diffusion Models, SIGGRAPH 2023
>
> [2] Linguistic Binding in Diffusion Models: Enhancing Attribute Correspondence through Attention Map Alignment, NeurIPS 2023
>
> [3] Expressive Text-to-Image Generation with Rich Text, ICCV 2023
>
> $^{1)}$ The problem we would like to solve is that pretrained large diffusion models are designed without considering any contextual bias which can be useful in multiple scenarios to control (c.f., L075 - L098 in the main paper). Thus, we suggest the way to obtain the contextual bias and the way to control it under the causal framework.
>
>
> > For this task, pre-storing 1TB of datasets seems excessively costly.
>
> After preprocessing, we can have the 1,130,195 samples of $p(C')$ (c.f., Eq. 7 in the main paper), which occupies only 81 MB. The end-users can simply leverage the precomputed $p(C')$ as we will release it, which means they do not need 1TB storage for running the preprocessing stage. We apologize for the confusion.

---

> ### Author Response · Authors · 2024-12-04
> **Second Response to Reviewer YF36 (2/2)**
>
> > It is essential for this work to demonstrate the necessity of this approach.
>
> - Wrapping up the necessities of our methods that are mentioned in our paper:
>
>   - As mentioned in L077-078 of our paper, CB+ could be useful if a given condition is not detailed enough to describe the whole scene. The generated sample can miss some objects that the user wanted to put together without directly mentioning, and CB+ can naturally autofill some visual components that have not been explicitly conditioned.
>
>   - As mentioned in L085-088, if non-trivial and diverse object combinations are desired from the generated images, CB- can be a solution. This is because we can extrapolate the object combinations of the generated sample beyond the co-occurrence statistics.
>
> - During the rebuttal, to more clarify the reviewer’s concern, we further provide the use-cases of CB- below:
>
> While contextual bias may not be an issue for all users, we contend that in creativity use-cases, e.g., inspiring or augmenting the creativity of content makers, contextual bias could be an issue because it limits the creativity of the images. Particularly, we suggest that there are naturally at least two distinct stages in creative content generation using diffusion models: (1) Ideation and (2) Refinement. In the ideation stage, the content maker may only have a vague topic in mind as encoded by a simple prompt as a starting point, such as "an astronaut". Our CB- approach can then generate many diverse feasible scenarios around this prompt. Our diverse scenarios can then inspire new ideas for the content creator, such as "An astronaut coming across a huge dinosaur/elephant on an alien planet" or "A space-born girl exploring the Milky Way" (Fig 3 in the Supplementary Material). After the ideation stage, the content creator can refine the idea by manually augmenting and updating the prompt to arrive at their final creation, such as "A space-born girl exploring the Milky Way. She is excited for the upcoming visit to Earth for the first time of her life". Our CB- approach is primarily useful for the ideation stage. Once an idea is selected, manual prompt design can be used to refine the final created image but this is not useful for generating new ideas.
>
> For example, suppose there is a person who needs creative thinking about a rabbit. This person could work in advertising companies, in the film or book industry, or could be a content creator. With existing sampling methods (Baseline 1 and 2 in Fig. 25 and 26 in the Supplementary Material), by prompting "a cute rabbit", it is not easy for him to get inspired and come up with interesting stories because most of the generated images are not diverse enough and very similar. With ours, however, a lot of interesting scenes can be generated, and he can easily come up with a lot of stories like "a musician rabbit playing the guitar", "a pet rabbit secretly watching a television when the owner is out", "an excited rabbit couple visiting a newly opened market in the town", "a rabbit traveling the forest wearing a backpack, slightly overwhelmed by an upcoming adventure", "a rabbit riding a mowing machine before inviting his fiance", "A rabbit comforting a girl crying in the living room", "A rabbit rowing a boat to tour the town", etc. After choosing one of them (e.g., "a musician rabbit playing the guitar"), he can add more details like "a musician rabbit wearing sunglasses and playing the guitar in the stadium with his band. He is playing rock music". As seen in this example, our proposed interventional sampling could be used to assist people to be creative. Once they get the idea, they can easily refine the prompt.

---

### Official Review · Reviewer_pXtd · 2024-11-01

**Soundness:** 2
**Presentation:** 2
**Contribution:** 2
**Rating:** 3
**Confidence:** 5

**Summary:**

This paper addresses the issue of contextual bias in image generation within the framework of causal inference. By leveraging this formulation, the authors propose two methods to either strengthen or weaken the influence of contextual biases during the image generation process. These methods rely on utilizing LLMs or VLMs to modify text prompts. The authors suggest that these adjustments will lead to more diverse and realistic generated results.

**Strengths:**

This paper provides an interesting formulation for the problem of contextual bias in image generations in the context of causal inference. This provides a novel perspective for thinking about how contextual bias influences generated images.

**Weaknesses:**

1.	While this paper formulates the problem of contextual biases in image generation using causal graphs and confounders, this formulation is overcomplicated and unnecessary for addressing the problem at hand. Although the theoretical framing is interesting, the proposed method largely boils down to a refined form of prompt engineering.
2.	The major concern for this paper is the novelty of the proposed method. Retrieving co-occurring objects using LLMs and identifying objects appearing in images using VLMs is trivial and has been commonly practiced in the task of text-to-image generation. The integration of LLMs and VLMs for prompt engineering is widely known and not innovative.
3.	Dealing with contextual biases in image generation is not a particularly challenging task for modern diffusion models like Stable Diffusion, DALLE, and Flux. These models are highly capable of generating diverse and complex images based on input prompts. They can easily generate unconventional combinations like “astronaut riding a horse on Mars” with prompt engineering along, without the need for special techniques to bypass contextual biases.
4.	The experiment should include more challenging cases that truly require causal modeling to demonstrate the significance of the approach. Without such cases, the relevance of the method remains limited.

**Questions:**

Please refer to the concerns raised in the weaknesses section above.

---

> ### Author Response · Authors · 2024-11-26
> **Response to Reviewer pXtd (1/2)**
>
> - (W1) **Formulation is overcomplicated and unnecessary for addressing the problem at hand.** In this paper, we explicitly model contextual bias using a causal framework. We emphasize that we aim for a principled and theoretically grounded approach to modeling contextual bias as opposed to a heuristic or ad-hoc approach. Thus, we first derive the appropriate equations from a principled viewpoint and then approximate these with foundation models. As opposed to the regular diffusion sampling method, our causal framework enables to strengthen the contextual bias (to assist to autofill the scene) or to weaken the contextual bias (to draw a novel scene with non-trivial object combinations). We thoroughly checked again our derivation and formulation and believe that our formulations in Eq. 3 and 7 are *not* overcomplicating nor unnecessary. We would appreciate it if the reviewer could provide a simpler but *principled*  way to address  the problem of contextual bias under a causal framework.
>
> - (W2) **Novelty of the proposed method ---Retrieving co-occurring objects using LLMs and identifying objects appearing in images using VLMs is trivial and has been commonly practiced in the task of text-to-image generation.** While individual steps in our method like retrieving co-occurring objects has likely been used before (though we would appreciate explicit references), we cannot find any work that uses these steps in a principled way to address contextual bias in a causal framework. Our novelty stems from how we derive and integrate these steps. The approximations we use via LLMs or VLMs are empirically justified by our strong results. Again, we emphasize that our novelty does not stem from the individual steps but from the principled derivation and specific combination and integration of those steps. If we have missed related works that develop a principled method for handling contextual bias, we would appreciate specific references so that we can improve our paper.
>
> - (W1) **The proposed methods boils down to a refined form of prompt engineering.** We respectfully disagree with both the implied claim that prompt engineering is not valuable and that our method is merely prompt engineering. First, we argue that prompt engineering itself has great value, and there are many highly cited research papers (e.g., [1-3]) directly related to prompt engineering. Thus, we believe your implicit assumption that prompt engineering is unworthy to be published is unfounded. Second, we argue that there are at least two independent components of our method. The first is the principled implementation-agnostic derivation of handling contextual bias using a specific form of interventional sampling. The second is the actual implementation of our interventional sampling by using prompt engineering. The mere fact that we can use prompt engineering as a way to approximate the necessary sampling should not be viewed as a weakness. To further emphasize the independence of the methodological contributions, we implemented a method for the conditional sampling, i.e., the conditional $p\_{\theta} \( X|y, C'=c' \)$, that does not involve prompt engineering. To be specific, since $y$ and $c'$ encode fundamentally different information, we can apply a sampling method Generalized Composable Diffusion Models (GCDM) [4] and  incorporate $y$ and $c'$ in the score space during the denoising process. By doing so, we can control the tradeoff between diversity due to $c'$ and the original prompt $y$. Please see the Additional experiment results 2 in "Response to all reviewers".
>
> [1] Language Models are Unsupervised Multitask Learners, 2019
>
> [2] Language models are few-shot learners, NeurIPS’20
>
> [3] Chain-of-Thought Prompting Elicits Reasoning in Large Language Models, NeurIPS’22
>
> [4] Enhanced Controllability of Diffusion Models via Feature Disentanglement and Realism-Enhanced Sampling Methods, ECCV'24

---

> ### Author Response · Authors · 2024-11-26
> **Response to Reviewer pXtd (2/2)**
>
> - (W3, W4) **Modern diffusion models can easily generate unconventional combinations like "astronaut riding a horse on Mars".** We heartily agree that modern diffusion models can easily generate unconventional combinations. Indeed, this assumption is the basis for our method and is exactly why we use pretrained diffusion models. Our goal is not to create a better diffusion model but to *automatically* control contextual bias. Or, to put it another way, our method can *automatically* generate creative and imaginative prompts. Your prompt "astronaut riding a horse on Mars" is rather a good example of why we need interventional sampling (Eq. 7 in the main paper). Let a prompt $y$ be "Astronaut on Mars" (already imaginative but not particularly creative) and a contextual bias $c'$ be "horse" and "man" (a very natural context). These seemingly unrelated $y$ and $c'$ combination successfully draws a creative and imaginative scene on top of the strong power of the modern diffusion models. However, Where did the prompt come from? A human came up with a prompt idea (astronaut on Mars) but then specifically chose words that would not usually co-occur even if they are natural in other contexts (i.e., man riding a horse). Yet, the human effort to create these prompts is not scalable. For example, suppose we need to make 1000 prompts to generate creative and novel scenes where an astronaut appears. We might be able to come up with 10-20 scenes quickly, but it is not necessarily easy to scale up. By leveraging our causally-grounded interventional sampling, however, we can generate thousands of novel scenes without any human labor because the retrieved contextual bias $c' \sim p(C')$ can extrapolate the co-occurring object combinations beyond data-driven correlation.

---

> > ### Comment · Reviewer_pXtd · 2024-11-27
> >
> > Thank you for your detailed response to my review. While I appreciate the clarification, I still have concerns regarding the core assumption that contextual bias is a significant issue for image generation using diffusion models.
> >
> > Firstly, I believe it is essential to provide concrete examples demonstrating that unwanted contextual bias actually arises in the images generated by current diffusion models. The presence of such bias should be clearly evidenced with specific cases, especially considering the capabilities of existing models like Stable Diffusion and DALLE 3. Without these examples, it is difficult to evaluate the magnitude of the problem your method is attempting to address.
> >
> > Additionally, without explicit comparisons in your experiments with baseline methods (such as simple prompt engineering or using negative prompts), I remain unconvinced that your sampling method offers a substantial advantage or is necessary.
> >
> > For instance, in your Figure 2, why not directly prompt “A living room with couches and a lamp on a beach” or “A white and brown spotted dog wearing a Santa Claus hat with water, rocks, grass in the background,” instead of employing your complex sampling method? These examples can easily be generated using basic prompting, which raises questions about the added value of your proposed approach. To make a stronger case for the necessity of your method, you need to demonstrate scenarios where it outperforms baseline methods—cases where simple prompting would fail.
> >
> > I fully agree that prompting techniques such as chain-of-thought can be valuable, particularly because they have been extensively validated as effective for solving problems that simple prompts cannot address. However, this level of validation and thorough comparison is not present in your paper. As such, I would need to see similar extensive verification of your method’s effectiveness to be convinced of its value.
> >
> > Furthermore, with regard to the idea of "automatically generating creative and imaginative prompts," this seems easily achievable with the use of LLMs, which are already capable of generating diverse prompts with minimal effort. Again, I would need to see a more direct and compelling comparison of your method’s effectiveness against baseline methods like prompting or prompting with negative prompts, both quantitatively and qualitatively, to justify the need for your approach.
> >
> > I hope these points clarify my concerns. I look forward to seeing further comparisons and more detailed evidence demonstrating the necessity and advantages of your proposed method.

---

> ### Author Response · Authors · 2024-12-03
> **Second Response to Reviewer pXtd (1/3)**
>
> We first address the core concern of when there is unwanted bias. Then we will provide extensive qualitative and quantitative results demonstrating that prompt engineering approaches including negative prompting cannot exhibit the creativity yet realism of our approach.
>
> > I still have concerns regarding the core assumption that contextual bias is a significant issue for image generation using diffusion models.
>
> While contextual bias may not be an issue for all users, we contend that in creativity use-cases, e.g., inspiring or augmenting the creativity of content makers, contextual bias could be an issue because it limits the creativity of the images. Particularly, we suggest that there are naturally at least two distinct stages in creative content generation using diffusion models: (1) Ideation and (2) Refinement. In the ideation stage, the content maker may only have a vague topic in mind as encoded by a simple prompt as a starting point, such as "an astronaut". Our CB- approach can then generate many diverse feasible scenarios around this prompt. Our diverse scenarios can then inspire new ideas for the content creator, such as "An astronaut coming across a huge dinosaur/elephant on an alien planet" or "A space-born girl exploring the Milky Way" (Fig 3 in the Supplementary Material). After the ideation stage, the content creator can refine the idea by manually augmenting and updating the prompt to arrive at their final creation, such as "A space-born girl exploring the Milky Way. She is excited for the upcoming visit to Earth for the first time of her life". Our CB- approach is primarily useful for the ideation stage. We totally acknowledge that once an idea is selected, manual prompt design can be used to refine the final created image but this is not useful for generating new ideas.
>
> For example, suppose there is a person who needs creative thinking about a rabbit. This person could work in advertising companies, in the film or book industry, or could be a content creator. With existing sampling methods (Baseline 1 and 2 in Fig. 25 and 26 in the Supplementary Material), by prompting "a cute rabbit", it is not easy for him to get inspired and come up with interesting stories because most of the generated images are not diverse enough and very similar. With ours, however, a lot of interesting scenes can be generated, and he can easily come up with a lot of stories like "a musician rabbit playing the guitar", "a pet rabbit secretly watching a television when the owner is out", "an excited rabbit couple visiting a newly opened market in the town", "a rabbit traveling the forest wearing a backpack, slightly overwhelmed by an upcoming adventure", "a rabbit riding a mowing machine before inviting his fiance", "A rabbit comforting a girl crying in the living room", "A rabbit rowing a boat to tour the town", etc. After choosing one of them (e.g., "a musician rabbit playing the guitar"), he can add more details like "a musician rabbit wearing sunglasses and playing the guitar in the stadium with his band. He is playing rock music". As seen in this example, our proposed interventional sampling could be used to assist people to be creative. Once they get the idea, they can easily refine the prompt, as the reviewer pointed out.

---

> ### Author Response · Authors · 2024-12-03
> **Second Response to Reviewer pXtd (2/3)**
>
> > I would need to see a more direct and compelling comparison of your method’s effectiveness against baseline methods like prompting or prompting with negative prompts, both quantitatively and qualitatively, to justify the need for your approach.
>
> - We conducted additional experiments with the suggested baselines to verify the advantage of our proposed interventional sampling over the existing methods. The results are reported both quantitatively and qualitatively.
> - Experiment settings:
>   - Baseline 1 is implemented with simple prompt engineering. Before being used as a condition to SDXL, a prompt is engineered by Gemini with a query of ``f"briefly modify the given prompt to be creative. Answer in one sentence.: {prompt}"``. The answer is used as a text input.
>   - Baseline 2 is implemented with a combination of prompt engineering and negative prompt techniques. A prompt is engineered by Gemini with a query of ``f"What would be the frequently co-occurring objects that can be likely placed in the scene generated by the given prompt ‘{prompt}’? Do not answer the words mentioned in the prompt. Answer 10 objects except for {important_obj} one line with comma."``, where ``important_obj`` is manually predefined as a key object from the prompt, e.g., ''ancient dragon'' from ''A fantasy illustration of ancient dragon''. Once the co-occurring objects are obtained, we use them as a negative prompt during the sampling process to remove contextual bias, as suggested by the reviewer.
>   - We use 13 queries. We randomly sample 20 samples per prompt (and per setting, e.g., baselines and ours) and show all of them without cherry picking.
> - Results: The image results are shown in the Supplementary Material. We can see that the results from our interventional sampling show better performance in creative scene generation than the baselines. For example, from Fig. 1,3, and 5, the Baseline 1 results such as an astronaut in space and photo of Mars and artistic autumn forest in diverse viewpoints could be considered creative. However, it is limited in generating diverse creative scenes as most of the images contain similar object compositions (c.f., the lowest LPIPS score in Table 3 below). We believe it is not suitable for helping and inspiring humans to be creative. In the case of Baseline 2, from Fig. 10 and 20, we can see that the negative prompt technique removes the contextual bias related to the prompt, e.g., the removed snow in Fig. 10 and 20. However, the diversity is still limited as it is not exactly modeling the interventional distribution from causal perspective, and thus the backdoor path in Fig. 3 in the main paper remains and negatively affects the image generation process. On the other hand, our results show impressive performance in generating diverse and creative scenes. For example, from Fig. 3, we can see an astronaut standing in front of a dinosaur, an astronaut walking through a street with debris, an astronaut standing next to huge elephants, an astronaut standing on the river, etc. (c.f., the highest LPIPS score in Table 3 below). Additionally, we also provide the qualitative prompt-engineered results in Table 4 below. Please let us know if the reviewer wants to see additional engineered prompts for the specific figures.

---

> ### Author Response · Authors · 2024-12-03
> **Second Response to Reviewer pXtd (3/3)**
>
> Table 3: Performance comparisons of new baselines, suggested by the reviewer.
> |     CB-        | Baseline 1 | Baseline 2 | ours |
> | :---------------- | :------: | :------: | :------: |
> | Diversity (LPIPS, &uarr;) | 0.6344 | 0.6465 | **0.6678** |
>
> Table 4: The engineered prompts by the baselines given a prompt of ``Photo of an astronaut''. (0,0) means the first row and the first column of the corresponding figures (i.e., Fig. 3 and Fig. 4 in the Supplementary Material).
> |             | Engineered prompt in Baseline 1 | Negative prompt in Baseline 2 |
> | :---------------- | :------: | :------: |
> | (0,0) | "Capture the ethereal grace of an astronaut floating through the vast expanse of the cosmos." | "Helmet, spacesuit, space shuttle, lunar module, EVA suit, oxygen tank, spacecraft, telescope, control panel, rover" |
> | (0,1) | "Astronaut, lost in the ethereal expanse, gazes into the cosmic abyss, a solitary sentinel amidst a celestial symphony." | "Spaceship, helmet, spacesuit, moon, stars, planets, telescope, satellite, Earth, rover" |
> | (0,2) | "A lonesome astronaut floats amidst a sea of stars, his visor reflecting the cosmic tapestry." | "Spaceship, spacesuit, helmet, gloves, oxygen tank, radio, mission patch, notebook, pen, camera" |
> | (0,3) | "A snapshot of an intrepid space traveler suspended amidst a sea of stars, capturing the vastness of the cosmos." | "Spaceship, helmet, Earth, stars, moon, spacesuit, gloves, boots, backpack, microphone" |
> | (0,4) | "An ethereal wanderer, suspended amidst celestial expanse, their suit shimmering with starlight" | "Spacesuit, helmet, oxygen tank, backpack, gloves, boots, lunar lander, rover, moon, stars" |
> | (1,0) | "Capture the boundless expanse through the lens of an astronaut's visor." | "Spaceship, helmet, gloves, suit, visor, jetpack, oxygen tank, boots, radio, camera" |
> | (1,1) | "A celestial explorer floats effortlessly amidst a cosmic tapestry, capturing the boundless depths of the universe." | "spaceship, planet, moon, star, helmet, gloves, oxygen tank, lunar rover, space suit, satellites" |
> | (1,2) | "Astronaut adrift in the cosmic abyss, a solitary sentinel amidst swirling nebulas and distant galaxies." | "Helmet, spacesuit, backpack, gloves, moon boot, lunar module, space shuttle, telescope, satellite, planet" |
> | (1,3) | "An ethereal portrait of a cosmic traveler, suspended against the starlit infinitude." | "Moon, space suit, helmet, spacecraft, rocket, stars, Earth, gloves, oxygen tank, flag" |
> | (1,4) | "An interstellar voyager suspended in the cosmic ballet, reaching for the unknown." | "spacesuit, helmet, gloves, spaceships, satellite, rockets, moon, earth, telescope, lunar module" |
> | (2,0) | "Astronaut floating through the enigmatic cosmos, adorned in a luminescent spacesuit, against a backdrop of shimmering stars." | "Spacesuit, gloves, helmet, space shuttle, lunar rover, stars, moon, planet, rocket, space station" |
> | (2,1) | "Immerse yourself in the awe-inspiring cosmos with a captivating image of an astronaut suspended in the boundless void." | "Space suit, helmet, moon, stars, spaceship, rocket, alien, flag, rover, space station" |
> | (2,2) | "Capture the ethereal beauty of a lone astronaut amidst the cosmic expanse, their solitude contrasting with the vastness of the universe." | "Helmet, spacesuit, gloves, boots, backpack, flag, spaceship, moon, stars, planet" |
> | (2,3) | "A solitary figure clad in a cosmic suit, adrift in a celestial tapestry of stars and nebulae." | "Rocket, space suit, telescope, space station, moon, stars, planets, spacecraft, satellite, flag" |
> | (2,4) | "Capture the cosmic dance of an astronaut frolicking amidst celestial bodies." | "spaceship, planet, stars, spacesuit, moon, helmet, gloves, boots, backpack, tools" |
> | (3,0) | "An ethereal voyager amidst a cosmic expanse, capturing the wonders of the unknown." | "Spaceship, space suit, helmet, oxygen backpack, gloves, boots, moon, stars, planets, Earth" |
> | (3,1) | "Here's a captivating photo of an astronaut floating effortlessly amidst a celestial tapestry of stars, their suit illuminated by the ethereal glow of distant galaxies." | "Spacecraft, spacesuit, helmet, backpack, gloves, visor, moon, stars, planet, flag" |
> | (3,2) | "Astronaut amidst the celestial tapestry, his gaze lost in the cosmic void." | "Moon, Spaceship, Earth, Rocket, Satellite, Spacesuit, Telescope, Planet, Stars, Space Shuttle" |
> | (3,3) | "Capture the ethereal solitude of an astronaut suspended amidst the celestial tapestry." | "spacesuit, moon, starship, planet, space station, spaceship, meteor, alien, spaceship, satellite" |
> | (3,4) | "A breathtaking capture of a lone astronaut floating amidst the celestial tapestry, their spacesuit illuminated against the backdrop of a vibrant interstellar void." | "Spaceship, spacewalk, helmet, visor, spacesuit, backpack, oxygen tank, gloves, boots, moon" |

---

### Official Review · Reviewer_aKEP · 2024-11-03

**Soundness:** 3
**Presentation:** 2
**Contribution:** 2
**Rating:** 6
**Confidence:** 4

**Summary:**

The authors propose two causally-motivated sampling frameworks for Latent Diffusion Models, which either increase or decrease their contextual biases.

1. CB+ increases the bias by using an LLM to describe confounders (objects) in a scene, and conditioning LDM sampling on these confounders
2. CB- decreases the bias by "retrieving" confounders, marginalized over the distribution of unconditionally generated images. This attempts to retrieve confounders (objects) which are not explicitly co-occuring with the original scene, thereby increasing scene diversity.

The authors present results on Visual Genome and use COCO to sample confounders.

**Strengths:**

1. This is a relevant and important problem for the community, and I found it well motivated. I appreciate the nuance in stating contextual bias is “not inherently bad” (L57-58) and providing two frameworks to tweak the bias in both directions
2. There are many experiments examining specific aspects of the framework, e.g. its impact on realism and diversity of generated images (Tab.1, 4), adherence to the original prompt (Tab 3), qualitative results (Fig 5), and its complementary nature with other frameworks (Fig 6)
3. The CB- framework is a very interesting contribution - if one is able to learn a "good" confounder distribution, it may help adding more diverse contextual biases to generative models

**Weaknesses:**

1.	Writing flow needs improvement. I found myself having to skip ahead to find where things were introduced or explained in the writing, and in some areas I was left with no clear answer (see below)
2.	What LDM is used for experiments (Fig 2, 4-6) ? Visually, it appears to be something similar to older versions of Stable Diffusion (e.g. 1.4, 2.1) From *L509*, it doesn’t seem to be SDXL throughout. This is very unclear and greatly detracts from being able to contextualize the results (some details in Weakness 3). Please explicitly add these details for all experiments.
3.	It appears that CB+ is replacing the contextual bias of the LDM with the contextual bias of the LLM (Gemini), and CB- is doing the same with a VLM (LlaVa). Assuming that the LDM is a slightly weaker model (see Weakness 2) detracts slightly from the FID comparisons Tab.1 – Gemini and LlaVA have: 1. much higher capacity (# params) 2. Much larger pretrain datasets than older Stable Diffusions, and thus their contextual biases may be of much higher quality. This makes it harder to make a fairer comparison. If all these results are with SDXL, a much stronger LDM, this is less of a concern (but this is unclear and should be specified)
4.	How the retrieved confounder $c’$ is used practically in CB+ and CB- is very unclear (I understand the math from Eq. 3 and Eq. 7). Are you adding the nouns from $c’$ directly to the prompt $y$ to generate a new image $x$? From *L387*, it sounds like you do not do this. My understanding of CB- (mostly from *L386-395*) is that you generate 10K images from COCO test+val captions and extract a set of nouns from all these images. You then randomly add these nouns to the prompt *y* (since you are not conditioning *y | c’*). In my opinion, this is the primary weakness of this work. Please provide a step-by-step description of how $c'$ is incorporated in the image generation process in practice for both CB+ and CB-.
5. The captions of figures need to be more self-contained. It is quite hard to understand them without referring back and forth from the text.

**Questions:**

1. I have a simpler baseline to suggest instead of the multi-step sampling chain in Eq. 6: use a VLM on the original $(x, y)$ as: "*given this image, describe a list of common nouns that do not occur in this scene, but could be reasonably expected to co-occur and increase the diversity*" – this will give you $c’$. You can also check for errors by asking the VLM to extract nouns from the scene (which you already do, *L270*) and the CB- technique (*L237-241*) and removing them if they are extracted from the above prompt. It is unclear to me if the sampling chain would outperform this baseline, especially because while marginalizing over truly unconditional samples as in Eq 6 *will* increase scene diversity, but can give you objects that are very out of place (e.g. Fig 4, a tree in a bedroom, motorcycles in a kitchen, etc.) This is merely an alternate suggestion, but I would like to hear the authors' thoughts about why this may or may not work compared to CB-.
2. I am a little confused by Eq 6. The starting point is marginalizing the likelihood of prompt $y$ given image $x’$ over all unconditionally generated images $x’$, which you do with a VLM. To compute this, you need to marginalize over *all* possible unconditionally generated images, which is intractable. A reasonable empirical approximation is to get a large collection of (hopefully diverse) unconditional images and marginalize over them, which is what I believe you are doing? **(a)** How many images $x’$ do you marginalize over? Is it $10000$ as you write in Fig. 1 and is this from COCO test+val (*L387*)? Please provide these details explicitly in Sec 3.3 **(b)** Are these diverse enough to get the non-co-occurring conditionings you need to reduce contextual bias? I suggest a small comment or discussion towards this question.
3. I would like more information on using CB+ and CB- with other conditionings (*L472-L483*), as I find this quite interesting and practically relevant for the community – a more explicit description of how alternate conditionings (e.g. ControlNet content or DEADiff style) can be used complementary to CB+ and CB- would be helpful

---

> ### Author Response · Authors · 2024-11-26
> **Response to Reviewer aKEP**
>
> - (W2) **Clarifications of the used models for the figures in the paper.** SD 2.1 is used for Fig. 2 and 4 and SD 1.5 is used for Fig. 6. SDXL is used for Fig. 7.
>
> - (W3) **FID results with SDXL for fair comparisons.** To resolve the concern of the reviewers about the fair comparisons, we conduct additional experiments to measure FID based on SDXL. Please see the Additional experiment results 1 in "Response to all reviewers".
>
> - (W4) **How is $c'$ incorporated in the image generation process?** In the main paper, to model $p\_{\theta} \( x|y, c' \)$, $c'$ is directly added in the prompt space. For example, if $c'$ is ``{chair, lamp, couch}`` and $y$ is ``A photo of a restroom``, the input condition is ``A photo of a restroom, chair, lamp, couch``. One step forward, during the rebuttal period, we explore a sampling method Generalized Composable Diffusion Models (GCDM) [1] to implement the CB- context conditioning in a different way than simple text concatenation. Please see the Additional experiment results 2 in "Response to all reviewers".
>
>   [1] Enhanced Controllability of Diffusion Models via Feature Disentanglement and Realism-Enhanced Sampling Methods, ECCV'24
>
> - (W4, Q2) **Clarifications on modeling $p(c')$** We apologize for the confusion. (Q2 (a)) As mentioned in L282-284, 1,130,195 samples are used to get $p(c')$. (Q2 (b)) Yes, empirically it is very diverse as more than one million of unconditionally generated images are used to retrieve $p(c')$. We further provide pseudo code for explaining the process more fully. Please see the Clarifications in "Response to all reviewers".
>
> - (Q1) **Discussion on the reviewer's suggestion and why it may or may not work compared to CB-.** The reviewer suggested another way to obtain $c'$ by asking VLM about "given this image, describe a list of common nouns that do not occur in this scene, but could be reasonably expected to co-occur and increase the diversity". This would correspond to something like sampling from the conditional $p(c'|x,q)$, where $q$ is the question and $x$ is the image. First of all, the input setting is different from ours because our task is text-guided image generation which means we do not have an image input $x$. Perhaps, you meant that we first generate an initial image from the original prompt $y$, then ask the VLM for new objects, and then regenerate with a new prompt. This could be approximately formalized as $p(c'|y) \approx \mathbb{E}_{x}\[p(c'|x,q)p(x|y)\]$. Note that $c'$ depends on $y$, which is specifically what our interventional distribution aims to avoid. Furthermore, it is unclear if this has any proper causal interpretation. While this heuristic approach may increase diversity, it does not provide a way to disentangle context in a principled way as our approach does. Please note that the interventional (that we model in the paper) can associate unrelated objects and generate creative scenes, e.g., a tree in a bedroom.
>
> - (Q3) **Implementation details about ControlNet and DEADiff**
> To implement Fig. 6, we add the contextual bias $c'$ and the text prompt $y$ in the prompt space. Since both ControlNet and DEADiff take a text prompt as input, we simply sample $c'$ from the precomputed $p(c')$, and add it to the prompt $y$. The new prompt (combining $c'$ and $y$) is then fed into the networks.

---

> ### Comment · Reviewer_aKEP · 2024-11-30
>
> I thank the authors for their detailed response to my review. Here are some follow-up questions from the rebuttal:
>
> R0. Can the authors please provide a clear indication of what was changed in the revision from the original submission? Without this, it is very time consuming to identify if previous weaknesses / questions have been addressed in the writeup.
>
> R1. The details of models used in all experiments and figures need to be clearly mentioned in the paper, not just in the rebuttal, unless I missed this in the revision (see R0).
>
> R2. I appreciate the additional experiments on FID with SDXL which extend previous results on a weaker diffusion model - this largely addresses W3. I would still like a discussion on the pros and cons of choosing other contextual biases (Llava and Gemini) to override and steer the existing contextual biases of the LDM.
>
> R3. The details of incorporating $c'$ are now clear to me from the rebuttal (text concatenation to the prompt), but this is still totally unclear from the main paper and needs to be included, unless I missed this in the revision. I appreciate the additional experiment with a different sampling method (GCDM) - my takeaway from these results is that boosting the contribution of retrieved confounder $c'$ during guidance increases diversity, which seems intuitively correct if $y$ and $c'$ encode different information about the image. I suggest including this experiment in the Appendix to further highlight this point.
>
> R4. What are the 1,130,195 samples used to learn p(C')? Since any added diversity depends on the diversity present in this data, it would be nice to have some details about what kind of data this is (how many classes? what kind of classes? do these samples overlap with pretraining distributions of any LDMs /Llava ?)
>
> R5. I remain unconvinced and slightly confused by the authors' response to Q1. In Eq 6, the sampling chain begins by conditioning on unconditionally generated images $x'$, which the authors describe with Llava to get $y'$, which is then used as conditioning to generate $x''$, which is again described by Llava to obtain objects $c'$. My question is, why not just use Llava's learned contextual bias to describe potentially co-occurring objects $c'$ from the original unconditionally generated image $x'$, instead of doing this entire sampling chain? Here, $c'$ depends only on $x'$ and not $y$, preserving the goal of your interventional distribution. Please correct me if I have misunderstood Eq 6 and this process.
>
> R6. It is not obvious to me that the proposed method has a "proper causal interpretation" from the paper. This would be more clear if we knew the distribution (i.e. training data) used to learn $c'$ and how it differed from the $y \to x$ mapping learned by the LDM. Additionally, I need more justification/discussion of the claim that $y$ and $c$ are truly causal of generated image $x$, and not just commonly co-occurring. In my understanding, $x$ and $y$ co-occur frequently in the pretraining distribution of the LDM, and $x$ and $c$ co-occur in the pretraining distribution of the LDM (unconditional) combined with the associations learned by Llava. Please correct me if I have missed something.
>
> In my view, the main paper is still lacking details and needs some work to improve clarity and readability. With the additional experiments, pseudocode, and clarifications provided in the rebuttal, I increase my score to a 7 (which unfortunately I cannot actually do). I would further increase my score to 8 if the missing details (R1 - R4) and my queries above (R5, R6) are addressed directly in the main paper. With this in mind, I will currently maintain my score.

---

> ### Author Response · Authors · 2024-12-04
> **Second Response to Reviewer aKEP (1/4)**
>
> > (R0, R1, R3) Paper revision
>
> We appreciate the reviewer for his/her kind advice to revise our paper. We agree with the reviewers’ points; 1) mentioning what SD versions are used per figure, 2) adding more contents/details to the captions for the figures to make them self contained, and 3) including the GCDM experiment to the Appendix. As it already has passed the deadline to directly update the pdf, we will thoroughly revise and update the pdf accordingly for the camera-ready version of our paper.
>
> > (R2)  I would still like a discussion on the pros and cons of choosing other contextual biases (Llava and Gemini) to override and steer the existing contextual biases of the LDM.
>
> - Pros: naively using the existing contextual biases of LDM is actually simply the regular diffusion sampling (i.e., Fig. 3 (b) in the main paper). To model $p(C’)$ though, we essentially leverage VLM (in the case of CB-) and LLM (in the case of CB+). By leveraging VLM or LLM, we could approximate contextual bias $p(C')$ and control them to strengthen/weaken the contextual bias effects.
>
> - Cons: $p(C’)$ is an approximation of $p(C)$, which means it is not exact $p(C)$. However, we would like to emphasize that the approximating step is essential as we do not have access to all the training data (as mentioned in L104-L110 in our paper). Specifically, if we can have access to all of the training data for the pretrained Diffusion Models (i.e., billions of image $x$, its paired caption $y$), and if we can manually label all of co-occurring objects $c$ for each image, we can obtain the samples from true $p(C)$ which is more accurate than our approximated $p(C’)$. Since it is infeasible in general, we approximate $p(C’)$ by leveraging LLM or VLM.
>
>
> > (R5) why not just use Llava's learned contextual bias to describe potentially co-occurring objects $c'$ from the original unconditionally generated image $x'$, instead of doing this entire sampling chain? Here, $c'$ depends only on $x'$ and not $y$, preserving the goal of your interventional distribution.
>
> What the reviewer suggested is one of the possible implementations of Eq. 7 in our main paper:
>
> $p(X|\text{do}(Y=y)) \approx \mathbb{E}_{c' \sim p(C')} \[ p\_{\theta} \( X|y, C'=c' \) \]$, where $ p(C') \approx \sum\_{x'} p\_{\phi} (C'=c'|X=x') p\_{\theta}(X=x')$. However, as mentioned in [1], we observe clearly degraded performance of unconditionally generated images from Diffusion Models, and thus we add one more step to get more accurate co-occurring objects information, i.e., get the prompt $y’$ first from the unconditional generation and next get the co-occurring objects $c’$ from conditionally generated images.
>
> [1] the second paragraph of the introduction, “Return of Unconditional Generation: A Self-supervised Representation Generation Method”, NeurIPS’24

---

> ### Author Response · Authors · 2024-12-04
> **Second Response to Reviewer aKEP (2/4)**
>
> > (R4) What are the 1,130,195 samples used to learn p(C')? Since any added diversity depends on the diversity present in this data, it would be nice to have some details about what kind of data this is (how many classes? what kind of classes? do these samples overlap with pretraining distributions of any LDMs /Llava ?)
>
> As specifically shown in the pseudo code in "Response to all reviewers", we obtain the 1,130,195 samples of $p(C')$ by following the derived sampling chain. Some examples of $c' ~ \sim p(C')$ could be ["car", "trees", "flowers"], ["couch", "chairs", "table", "kitchen appliances", "island"], ["car", "cow", "man", "dog"], and ["bench", "trees", "people", "car"], where each list contains actually co-occurring objects of diffusion models.
>
> As for the details, we provide the highest and the lowest 100 object count matrices over the 1,130,195 samples in Table 3 and 4. The example for how to count each object is that, given the examples above, the count of "car" is 3 and the count of "trees" is 2. The vocab size (i.e., # of words) is 48774.
>
> As for the overlap with pretrained LDMs and LLaVA, since LDMs and LLaVA are leveraged as core steps to get the samples for $p(C')$ in our proposed sampling chain (c.f., Eq. (6) of our paper, and the pseudo code in the Response to all reviewers), we believe the pretrained knowledge of LDMs and LLaVA is the fundamental ingredient of the retrieved $p(C')$.
>
> > (R6) It is not obvious to me that the proposed method has a "proper causal interpretation" from the paper. This would be more clear if we knew the distribution (i.e. training data) used to learn $c′$ and how it differed from the
> $y \rightarrow x$ mapping learned by the LDM. Additionally, I need more justification/discussion of the claim that $y$ and $c$ are truly causal of generated image $x$, and not just commonly co-occurring. In my understanding, $x$ and $y$ co-occur frequently in the pretraining distribution of the LDM, and $x$ and $c$ co-occur in the pretraining distribution of the LDM (unconditional) combined with the associations learned by Llava. Please correct me if I have missed something.
>
> We first would like to define the meaning of each variable with some examples. We next describe the causal relationships between the variables. (c.f., Section 3.1 in the main paper)
>
>  $Y$ is text prompt and $X$ is image. $C$ is an unobserved confounding variable that we would like to approximate by $C'$. In our paper, the confounder $C$ indicates co-occurring objects statistics in our visual world. Suppose we have some paired samples $(Y=y,X=x,C=c)$ for each variable. Some examples would be ($y=$''Photo of street'', $c=$["car", "people", "traffic light", "traffic sign"], $x_{\text{street}}$), where $x_{\text{street}}$ is a corresponding street image,  and ($y=$''Photo of living room'', $c=$ ["couch", "lamp", "rug", "window", "plant"], $x_{\text{living room}}$).
>
>  As mentioned in [2], conceptually, the object co-occurrence statistics $C$ could affect either $X$ or $Y$ because $X$ and $Y$ are realizations of the visual world in different modalities (i.e., image and text). For example, there could be a tendency that "couch", "lamp", "rug", and "window" typically co-occur in the "living room", which could affect the data generation process. In other words, most living room images or most image descriptions with a word "living room" could keep containing the frequently co-occurring objects ("couch", "lamp", "rug", and "window"). These co-occurrence statistics implicitly but likely make the pretrained diffusion models generate an image with ("couch", "lamp", "rug", and "window") given a prompt "living room". Thus, our proposed causal graph has the edges $C \rightarrow X$ and $C \rightarrow Y$. The edge $Y \rightarrow X$ comes from the pretrained text-guided diffusion models that take $Y$ as an input and output $X$. (c.f., Fig. 3 in the main paper)
>
> Now, we retrieve a confounder $C'$ by approximating the inaccessible hidden confounder $C$ and control it following CB+ and CB- (c.f., Fig. 3 (a), (c) in our paper).
>
> [2] Fig. 5, Visual Commonsense R-CNN, CVPR 2020

---

> ### Author Response · Authors · 2024-12-04
> **Second Response to Reviewer aKEP (3/4)**
>
> Table 3: Objects with maximum counts over 1,130,195 samples of $p(C')$.
> | idx |    object       |   count    | idx |    object       |   count    |idx |    object       |   count    |
> | :------: | :------: | :------: | :------: | :------: | :------: |  :------: | :------: | :------: |
> | 1 | trees | 198471 | 35 | bridge | 22906 | 69 | sink | 13150 |
> | 2 | people | 177889 | 36 | boats | 22763 | 70 | bicycle | 13097 |
> | 3 | buildings | 165226 | 37 | umbrella | 22639 | 71 | mountains | 13056 |
> | 4 | table | 88523 | 38 | stairs | 22439 | 72 | doors | 12693 |
> | 5 | flowers | 87732 | 39 | person | 22256 | 73 | pool | 12633 |
> | 6 | chairs | 67823 | 40 | fence | 21442 | 74 | clouds | 12424 |
> | 7 | tree | 67083 | 41 | houses | 21201 | 75 | curtains | 12403 |
> | 8 | window | 63814 | 42 | sidewalk | 20855 | 76 | roads | 12316 |
> | 9 | door | 60191 | 43 | boat | 20602 | 77 | plates | 12153 |
> | 10 | water | 59227 | 44 | plant | 20297 | 78 | cups | 12064 |
> | 11 | building | 58941 | 45 | ceiling | 20170 | 79 | bushes | 12054 |
> | 12 | cars | 58103 | 46 | debris | 19821 | 80 | spoon | 12037 |
> | 13 | rocks | 56005 | 47 | bird | 19319 | 81 | snow | 11876 |
> | 14 | man | 54870 | 48 | umbrellas | 19208 | 82 | dog | 11859 |
> | 15 | chair | 54842 | 49 | shoes | 18944 | 83 | waterfall | 11240 |
> | 16 | house | 52547 | 50 | birds | 17811 | 84 | pillows | 11061 |
> | 17 | windows | 45746 | 51 | statue | 17634 | 85 | plate | 11013 |
> | 18 | grass | 44825 | 52 | chandelier | 17347 | 86 | desk | 10722 |
> | 19 | couch | 43895 | 53 | hat | 17144 | 87 | boxes | 10712 |
> | 20 | woman | 43746 | 54 | tables | 16591 | 88 | bottles | 10669 |
> | 21 | sky | 40773 | 55 | floor | 16519 | 89 | ottoman | 10455 |
> | 22 | books | 39953 | 56 | lights | 16126 | 90 | road | 10260 |
> | 23 | vase | 39135 | 57 | bowls | 15723 | 91 | animals | 10236 |
> | 24 | plants | 38752 | 58 | potted plants | 15466 | 92 | columns | 10231 |
> | 25 | lamp | 37621 | 59 | path | 14861 | 93 | leaves | 10194 |
> | 26 | rug | 36959 | 60 | vases | 14561 | 94 | brick wall | 10139 |
> | 27 | wall | 35135 | 61 | fish | 14173 | 95 | pots | 9894 |
> | 28 | mirror | 30589 | 62 | shelves | 14164 | 96 | fork | 9881 |
> | 29 | street | 30400 | 63 | trash | 14159 | 97 | motorcycle | 9799 |
> | 30 | bench | 28662 | 64 | fireplace | 14157 | 98 | coffee table | 9766 |
> | 31 | car | 27213 | 65 | bed | 14095 | 99 | television | 9724 |
> | 32 | bowl | 26184 | 66 | signs | 13993 | 100 | papers | 9678 |
> | 33 | clock | 25031 | 67 | river | 13704 | 101 | knife | 9436 |
> | 34 | potted plant | 24318 | 68 | scissors | 13228 | 102 | sign | 9340 |

---

> ### Author Response · Authors · 2024-12-04
> **Second Response to Reviewer aKEP (4/4)**
>
> Table 4: Objects with minimum counts over 1,130,195 samples of $p(C')$, thresholded by 100.
> | idx |    object       |   count    | idx |    object       |   count    |idx |    object       |   count    |
>  | :------: | :------: | :------: | :------: | :------: | :------: |  :------: | :------: | :------: |
> | 1 | waste | 115 | 35 | maze | 109 | 69 | plum | 105 |
> | 2 | vice | 115 | 36 | traffic markings | 109 | 70 | roller coasters | 104 |
> | 3 | white wall | 114 | 37 | no objects | 109 | 71 | fallen tree | 104 |
> | 4 | shakers | 114 | 38 | lampposts | 109 | 72 | wine bottles | 104 |
> | 5 | saws | 113 | 39 | door mail slot | 109 | 73 | bug | 103 |
> | 6 | trays | 113 | 40 | salad dressing | 108 | 74 | stoves | 103 |
> | 7 | stadiums | 113 | 41 | fish tank | 108 | 75 | hat stand | 103 |
> | 8 | eiffel tower | 112 | 42 | hairdryer | 108 | 76 | pendants | 103 |
> | 9 | ceiling beam | 112 | 43 | thermometer | 108 | 77 | nail polish brush | 103 |
> | 10 | cheese grater | 112 | 44 | fixtures | 108 | 78 | lamp base | 103 |
> | 11 | purple cabbage | 112 | 45 | drum | 108 | 79 | cheek | 103 |
> | 12 | wicker table | 112 | 46 | ferns | 108 | 80 | dip | 103 |
> | 13 | cake stand | 112 | 47 | other knick knacks | 107 | 81 | blue wall | 103 |
> | 14 | displays | 112 | 48 | kitchenware | 107 | 82 | drills | 102 |
> | 15 | other furniture | 112 | 49 | podium | 107 | 83 | red curtains | 102 |
> | 16 | end table | 112 | 50 | tapestry | 107 | 84 | coat rack | 102 |
> | 17 | dryer | 111 | 51 | parking ramp | 107 | 85 | fire truck | 102 |
> | 18 | sea | 111 | 52 | pocket watch | 107 | 86 | stone pillar | 102 |
> | 19 | ceilings | 111 | 53 | miniature accessories | 107 | 87 | measuring spoon | 102 |
> | 20 | floor rug | 111 | 54 | shutter | 107 | 88 | poster frame | 102 |
> | 21 | shades | 111 | 55 | chargers | 107 | 89 | bus stop | 102 |
> | 22 | soccer field | 111 | 56 | dollhouse clothes | 107 | 90 | bedding | 102 |
> | 23 | sunflower | 111 | 57 | electrical outlet | 107 | 91 | flower pot | 101 |
> | 24 | window shutter | 110 | 58 | sand dunes | 106 | 92 | walnuts | 101 |
> | 25 | driveways | 110 | 59 | fashion | 106 | 93 | religious vestments | 101 |
> | 26 | brick pathway | 110 | 60 | light bulbs | 106 | 94 | store window | 101 |
> | 27 | pottery | 110 | 61 | collages | 106 | 95 | paths | 101 |
> | 28 | a street | 110 | 62 | tissue box | 106 | 96 | carriages | 101 |
> | 29 | snowmen | 110 | 63 | dessert | 105 | 97 | diamond | 101 |
> | 30 | playgrounds | 110 | 64 | a rug | 105 | 98 | symbols | 101 |
> | 31 | polka dots | 110 | 65 | shrimp | 105 | 99 | rifle | 101 |
> | 32 | pearls | 110 | 66 | oven mitt | 105 | 100 | metal parts | 101 |
> | 33 | space shuttle | 109 | 67 | red hat | 105 | 101 | bell pepper | 101 |
> | 34 | window seat | 109 | 68 | other materials | 105 | 102 | mall | 101 |

---

### Official Review · Reviewer_p42H · 2024-11-04

**Soundness:** 3
**Presentation:** 3
**Contribution:** 3
**Rating:** 6
**Confidence:** 3

**Summary:**

This paper introduces a causally motivated approach to enhance image diversity and fidelity in large-scale diffusion models by addressing contextual bias, without the need for retraining or extensive data access. The proposed methods involve causality-inspired techniques to modulate the influence of contextual information during the diffusion process, thus balancing realistic image generation with diverse outputs. Through experiments on datasets like Visual Genome and COCO, the approach demonstrates significant improvements in metrics such as FID and LPIPS compared to standard diffusion models. This work contributes a novel framework for controlled image synthesis, enabling broader applicability of diffusion models in creative and diverse image generation tasks.

**Strengths:**

1. The paper introduces a novel, causally motivated approach to address contextual bias in diffusion models, which effectively enhances image diversity and fidelity without requiring retraining or extensive data.

2. The proposed methods are validated on multiple large-scale datasets, such as Visual Genome and COCO, demonstrating consistent performance improvements in key metrics like FID and LPIPS.

3. The framework is adaptable and efficiently addresses contextual bias within the diffusion process, broadening the application scope of diffusion models for diverse and controlled image generation.

**Weaknesses:**

1. There is a lack of robustness when the sampled confounder $𝐶′$ is semantically distant from the prompt $𝑌$, leading to generated images that may ignore the confounder altogether​. Besides, the framework’s dependence on predefined confounders may limit its flexibility when generating images outside of commonly biased contexts, reducing adaptability in less standardized environments.
2. The approach depends on complex causal graphs and sampling chains, which may lead to higher computational demands and slower generation times, limiting its scalability​.
3. Some generated images may exhibit unnatural object combinations, particularly when weakening contextual bias, which might detract from the realism of the results​.
4. While the framework introduces techniques to adjust contextual bias, it does not provide a quantitative evaluation of how well these adjustments meet specific user-defined objectives or bias levels.

**Questions:**

see weaknesses

---

> ### Author Response · Authors · 2024-11-26
> **Response to Reviewer p42H**
>
> - (W1-1) **Generated images can ignore the confounder.** As we mentioned in Limitations from the main paper, the phenomenon that diffusion models could ignore either one of $C'$ of $Y$ is observed, if they are semantically too far. However, we believe this is not a fundamental issue of our proposed sampling methods but rather a limitation of current diffusion models. As the power of diffusion models increases, this phenomenon would be reduced.
>
> - (W1-2) **The framework’s dependence on predefined confounders may limit its flexibility.** Thank you for pointing out a valid point. Yes, our predefined $p(C')$ might be limited in representing all the possible object-cooccurrence information. However, it contains fairly rich information as $p(C')$ is precomputed over 1130195 samples. Moreover, based on Eq. 7 in the main paper (i.e., $\mathbb{E}_{c'} \[ p\_{\theta} \( X|y, C'=c' \) \]$), our interventional sampling method generates an image by combining $y$ with $c'$, which means the generated images contain both $c'$ and $y$, mitigating the limitation of $p(C')$.
>
> - (W2) It is true that the derived sampling chain in Eq. 6 of the main paper is expensive. To make it up, as mentioned in L282-L287 of the main paper, we precompute $p(C')$, and this precomputation significantly reduces the sampling time. Without precomputing, it takes 12 seconds to sample $c'$ while it takes less than 1 second to sample $c'$ from the precomputed $p(C')$. Thus, we can confirm that the sampling cost of $c'$ does not affect the scalability/adaptability of our interventional sampling method.
>
> - (W3) Interestingly, our experiment results show that FID from CB- (weakening contextual bias) is consistently the best. Table 1 in the main paper is based on SD 2.1 and Table 1 shown in the "Response to all reviewers" is based on SDXL. We believe this is because FID measures both quality and diversity, and thus increasing diversity appropriately can improve FID.

---

> > ### Comment · Reviewer_p42H · 2024-11-27
> >
> > I will keep my rating since my score is one of the highest.

---

### Author Response · Authors · 2024-11-26
**Response to all reviewers (2/2)**

- ### Clarifications: Pseudo code for fully explaining how to retrieve the contextual bias $p(C')$.

- (**Reviewer aKEP, Reviewer YF36**) CB-: $p(X|\text{do}(Y=y)) \approx \textstyle \sum_{c'} p_{\theta}(X|Y=y, C'=c') p(C'=c')$, where $p(C’=c’) \approx \sum_{y'}\sum_{x''}p_\phi(C'=c'|X=x'') p_\theta(X=x''|Y=y')\sum_{x'}p_{\phi}(Y=y'|X=x')p_{\theta}(X=x')$

    `pretrained_diffusion_models`: used for both 1. unconditional generation $p_\theta(X=x')$ and 2. text conditional generation $p_\theta(X|Y=y, C'=c')$

    `pretrained_vlm`: used for implementing both terms $p_{\phi}(Y=y'|X=x')$ and $p_\phi(C'=c'|X=x'')$

    `preprocessing_function`: 1. To remove duplicated objects, 2. To filter out some exceptions that do not fit in the given answer form, 3.(possibly) To detect and remove inappropriate words.

    `y`: text prompt

    `k` : the number of objects (we use 10)

    `n_samples` : 1130195

    - #### Precomputing contextual prior: to retrieve the contextual bias that is implicitly learned during the pretraining stage.
      1. unconditional image generation $p_{\theta}(X=x')$
      ```
      def uncond_generation(pretrained_diffusion_models):
        for i in range(n_samples):
          uncond_x = pretrained_diffusion_models()
          uncond_x.save(f"uncond_{i}.png")
      ```
      2. obtaining empirical conditional $p_{\phi}(Y=y'|X=x')$
      ```
      def get_y_from_uncond(pretrained_vlm):
        query = "Shortly describe the scene in one sentence"
        for i in range(n_samples):
          uncond_x = image_load(f"uncond_{i}.png")
          uncond_caption = pretrained_vlm(image=uncond_x, prompt=query)
          uncond_caption.save(f"uncond_{i}.txt")
      ```
      3. conditional image generation $p_\theta(X=x''|Y=y')$
      ```
      def cond_generation(pretrained_diffusion_models):
        for i in range(n_samples):
          uncond_caption = text_load(f"uncond_{i}.txt")
          cond_x = pretrained_diffusion_models(prompt = uncond_caption)
          cond_x.save(f"cond_{i}.png")
      ```
      4. obtaining empirical contextual bias $p(C'=c')$ by computing $p_\phi(C'=c'|X=x'')$
      ```
      def get_contextual_bias(pretrained_vlm, preprocessing_function):
        query = "What objects are in the image? Answer {k} objects in English in one line with comma."
        for i in range(n_samples):
          cond_x = image_load(f"cond_{i}.png")
          cb_prior_sample = pretrained_vlm(image=cond_x, prompt=query)
          cb_prior_sample = preprocessing_function(cb_prior_sample)
          cb_prior_sample.save(f"cb_prior_sample_{i}.txt")
      ```

    - #### Now, we can sample from the mixture $\sum_{c'} p_{\theta}(X|Y=y, C'=c') p(C'=c')$ because we can sample from the precomputed p(C'=c').
      ```
      def cb_minus(y, cb_prior, pretrained_diffusion_models):
        cb = randomly_sample(cb_prior) # already preprocessed in d-step above.
        x = pretrained_diffusion_models(prompt = y + cb)

        return x
      ```
    ---

---

### Author Response · Authors · 2024-11-26
**Response to all reviewers (1/2)**

- We thank all the reviewers for sharing their valuable comments. We hope that our response can resolve reviewer's concerns.
- ### Additional experiment results:
  **1. (Reviewer aKEP, Reviewer p42H)** FID comparisons with SDXL. We conduct additional experiment to show that the performance of our proposed methods are consistent even with the stronger version of StableDiffusion. We sample 1000 images for each sampling method; CB+, Regular sampling (Reg), and CB-. SDXL-turbo is used to speed up the sampling time. The experiments are done with various settings such as VG (k) where k object words are used to generate an image, and COCO where a caption is used. The results are shown in Table 1 below. The observed pattern is almost identical with Table 1 in the main paper (with SD 2.1); CB- consistently shows better FID than both Reg and CB+ in all of the settings because it can diversify the generated objects by leveraging the retrieved confounder. CB+ also shows a similar pattern; better FID than Reg when a few words are given. This empirically shows that our proposed sampling methods with the stronger Diffusion Models (SDXL) can be as effective as with the weaker versions (SD 1.5 and 2.1).

Table 1: FID comparisons with SDXL. The higher FID scores than the main paper are because of the lower number of samples (1000) due to the limited time window of rebuttal.
|     FID (&darr;)          | CB+ | Reg | CB- |
| :---------------- | :------: | :------: | :------: |
| VG (1)        | *121.45* | 163.14 | **95.60** |
| VG (3)           | 121.41 | 114.69 | **94.10** |
| VG (5)    | 121.30 | 103.29 | **92.70** |
| VG (7)    | 110.99 | 100.57 | **94.24** |
| COCO | 66.93 | 67.42 | **61.86** |


  **2. (Reviewer aKEP, Reviewer pXtd)** During the rebuttal period, we explore a sampling method Generalized Composable Diffusion Models (GCDM) [1] to implement the CB- context conditioning in a different way than simple text concatenation. Since $y$ and $c'$ encode fundamentally different information, we can apply GCDM and incorporate $y$ and $c'$ in the score space during the denoising process. By doing so, we can control the tradeoff between diversity due to $c'$ and the original prompt $y$.
    - GCDM formulation: $p \( x|y, c' \) \approx \sum_{t=1}^T \[ \lambda_{y,c'} \nabla_{x_t} \log p\_{\theta} \(x | y,c',t\) +  \lambda_{y} \nabla_{x_t} \log p\_{\theta} \( x|y,t \) + \lambda_{c'} \nabla_{x_t} \log p\_{\theta} \( x|c',t \) + \nabla_{x_t} \log p\_{\theta} \(x,t \) \],$ where $t$ is timestep and $\lambda$'s are hyperparameters determining the strength of each guidance term.
    - Experiment setting: we use the first 500 captions of COCO testset. We generate 5 samples per caption (2500 samples in total) by using CB-, and compute the pair-wise perceptual distance by LPIPS (i.e., $_{5}C_2$). To compute the CLIP score, we obtain the objects information from the generated images by using LLaVA. We then compare the CLIP text-text similarity between prompt $y$ and the obtained objects.
    - Experiment results: Table 2 below shows the results. We can see that by introducing GCDM, we can control how much CB- adjusts regular sampling to get higher diversity in exchange for lower $y$ preservation.

[1] Enhanced Controllability of Diffusion Models via Feature Disentanglement and Realism-Enhanced Sampling Methods, ECCV'24


Table 2: Performance comparisons of various sampling settings for CB-. The used hyperparameters are denoted as GCDM ($\lambda_{y,c'}, \lambda_{y}, \lambda_{c'}$).
|     CB-        | Reg, GCDM (0, 1, 0) | GCDM (0.5, 1, 0) | Prompt engineering, GCDM (1, 0, 0) | GCDM (0.5, 0, 1)|
| :---------------- | :------: | :------: | :------: | :------: |
| Diversity (LPIPS, &uarr;) | 0.7095 | 0.7175 | 0.7385 | **0.7443** |
| Preserving $y$ (CLIP text, &uarr;) | **0.6927** | 0.6825 | 0.6471 | 0.5823 |

---

### Meta-Review · Area_Chair_jZpS · 2024-12-20

**Metareview:**

This paper addresses the issue of contextual biases in diffusion models.  By applying causal learning framework, the influence of confounding factors is either enhanced or mitigated. Specifically, the authors utilize LLMs/VLMs to sample and estimate the distribution of confounding factors. The idea proposed in the paper is interesting. However, one some limitations include complexity of the approach, scalability concerns and a good justification of why this approach is needed in the first place over prompt updating by LLMs.

**Additional Comments On Reviewer Discussion:**

While the reviewers agree that the idea is interesting, there are several main concerns that were raises
1. Robustness and scalability: There are concerns about the robustness of the approach, and if the approach can be scalable.
2. Modern diffusion models are good at generating novel combinations if they are specified in the prompt. While it is true that this approach can mitigate contextual biases for a given prompt, pXtd raised an interesting comment that the prompts can be modified by LLMs easily to generate novel combinations. It seems like a much more scalable approach, and it is unclear if the approach presented in this paper is better than that.
3. Hallucinations in LLMs leading to incorrect probability estimate is another issue.

Based on these discussions, I think the approach, while being novel, seems more complex than a simple LLM based prompt updating. And a strong comparison with such approach can further strengthen the paper. I regret to say that the paper in the current state is not ready for publication, and encourage the authors to improve the writing and come up with strong experiments to justify the imporance of mitigating contextual biases this way.

---

### Decision · Program_Chairs · 2025-01-22

Reject